# Research on Multi-Time Scale Optimization Strategy of Cold-Thermal-Electric Integrated Energy System Considering Feasible Interval of System Load Rate

**Bin Ouyang [1,2,\*], Zhichang Yuan [2]**  **, Chao Lu [2], Lu Qu [2] and Dongdong Li [1]**

1   Electric Power Engineering, Shanghai University of Electric Power, Shanghai 200090, China
2   Department of Electrical Engineer, Tsinghua University, Beijing 100084, China
\*   Correspondence: Oyang2014@163.com; Tel.: +86-188-1722-2035

**Abstract:** The integrated energy system coupling multi-type energy production terminal to realize multi-energy complementarity and energy ladder utilization is of great significance to alleviate the existing energy production crisis and reduce environmental pollution. In this paper, the topology of the cold-thermal-electricity integrated energy system is built, and the decoupling method is adopted to analyze the feasible interval of load rate under the strong coupling condition, so as to ensure the "source-load" power balance of the system. Establishing a multi-objective optimization function with the lowest system economic operation and pollution gas emission, considering the attribute differences and energy scheduling characteristics of different energy sources of cold, heat and electricity, and adopting different time scales to optimize the operation of the three energy sources of cold, heat and electricity, wherein the operation periods of electric energy, heat energy and cold energy are respectively 15 min, 30 min and 1 h; The multi-objective problem is solved by standard linear weighting method. Finally, the mixed integer nonlinear programming model is calculated by LINGO solver. In the numerical simulation, the hotel summer front load parameters of Zhangjiakou, China are selected for simulation and compared with a single time scale system. The simulation results show that the multi-time scale system reduces the economic operation cost by 15.6% and the pollution gas emission by 22.3% compared with the single time scale system, it also has a wider feasible range of load rate, flexible time allocation, and complementary energy selection.

**Keywords:** cold-thermal-electricity integrated energy system; multi-energy complementation; feasible interval of load rate; multi-objective; multi time scales; optimize operation

---

## 1. Introduction

As the driving force for social development and the indispensable whole of daily life, energy plays an increasingly important role in modern life. As for the problems that the traditional fossil energy is exhausted, the energy utilization efficiency is low and the environmental pollution is serious which people have to seek an efficient, energy-saving, environment-friendly and renewable energy production and utilization mode. At the same time, due to the differences and limitations in the development of different energy systems, the planning, design, operation and control of each energy system are often separated and lack coordination with each other [1]. As a result, the overall energy utilization rate of the system is low, the energy supply of the system is safe, and the self-healing ability is not strong [2]. Therefore, Integrated Energy Systems (IES) [3–6] that couple multiple energy production and consumption modes, absorb a large amount of renewable energy, realize multi-energy complementation, and improve the overall energy utilization rate should emerge.

In recent years, many scholars have paid close attention to the research and application of integrated energy system. In Europe, the University of Manchester in the United Kingdom took the lead in

developing a local integrated energy system, which integrates electricity/gas/thermal energy systems and a user interaction platform. The platform realizes three major functions: energy consumption mode, energy conservation strategy and demand-side response [2]; Denmark is the first country in the world to set a goal of completely separating from fossil fuels. It is expected to use 100% renewable energy by 2050, with emphasis on integrating different energy systems [7]; The University of Aachen and the German Federal Ministry of Economy and Environment launched the E-Energy Project [8] through demand-side response, achieving a high degree of integration of energy, information and capital, promoting the automation of energy service management, and successfully landing in Langenfel, Germany. The EU has set a target of carbon pollution from electricity production by 2050 [9], planned a new route for the EU power grid plan, and is committed to building a trans-European high-efficiency energy system by integrating the energy systems of various countries. In 2001, the U.S. Department of Energy put forward a comprehensive energy system development plan [10] aimed at improving the supply and utilization of clean and renewable energy and further improving the economy and reliability of the energy supply system. Japan established the Japan-Wireless City Alliance in 2010 [11] and is committed to the research of wireless city technology and the demonstration of the national integrated energy system. In China, a number of integrated energy system demonstration projects have been launched. Guangzhou Mingzhu Industrial Park, in combination with the future development direction and technical requirements of the city's power grid [12] that actively building an intelligent industrial park demonstration park for large-scale local consumption of renewable energy through cold/heat/electricity/gas multi-system coupling. Zhangbei wind/Light/Heat/Storage/Transmission Multi-energy Complementary Demonstration Project [13] in Zhangjiakou, Hebei province sets a precedent for large-scale multi-energy complementary power generation by comprehensively utilizing various energy storage and photo-thermal power generation technologies in order to build a strong smart grid. At the same time, the integrated energy system is also facing many difficulties and challenges in planning, modeling and optimizing operation.

In terms of optimal operation, document [14] establishes an economic optimal operation model of distributed energy system compatible with demand-side regulation, fully considers the cold/heat/electricity load in the distributed energy system which proposes quantum fireworks algorithm to solve the model. Literature [15] takes the cold/heat demand of 433 buildings as a model, compares the adaptability of two operation modes of heat determination and electricity determination in buildings, and finally points out that heat determination is not economical due to the high investment cost in the early stage, and the optimization of primary energy saving rate can improve the economic benefits of the factory. However, with the method of heat determination by electricity, when the demand for heat energy is strong, auxiliary equipment is needed to ensure the supply of heat energy. Excess heat is easily dissipated in the environment and wasted when the heat supply is excessive, so it is not ideal. In view of the irrationality of the methods of determining electricity by heat and determining heat by electricity, document [16] proposes energy scheduling algorithms aiming at minimizing operating cost, primary energy consumption (PEC) and carbon dioxide emission (CDE), respectively, points out that the three optimization algorithms do not have a common trend under common circumstances, and points out that the installation and operation of the system should be considered as long as PEC and CDE meet the standard requirements. However, using this method, different buildings need different analysis and evaluation to determine the optimal operating conditions. Literature [17] takes the minimum total operating cost as the optimization objective to maximize the utilization rate of resources. But, the problems mentioned above are all aimed at the single time scale operation of the system, and there are few researches on the multi-time scale characteristics of the integrated energy system. Documents [18] and [19] aim at the defects existing in the operation of the integrated energy system before the day, and use the two time scales of day-ahead load state estimation and real-time rolling correction to realize the optimization of real-time scheduling. Literature [20] considers the dynamic response characteristics of multi-time scale and multi-energy coupling systems, and starts with the modeling direction, constructs a network model for the combined calculation of electric/gas/heat/cold

multi-energy flows from both steady and dynamic aspects. In the process of solving the dynamic model, a hybrid stepping time domain simulation method for electromechanical transient simulation of power system and a medium and long-term transient simulation for non-power system are proposed. Finally, the electric heating coupling network is constructed for simulation verification. Document [21] proposes a multi-time scale multi-objective optimal joint scheduling model, which takes into account the different time scale characteristics of wind/water resources, electric/thermal load characteristics, peak load regulation capability, thermoelectric coupling characteristics of different thermal units, transmission capacity and system rotation standby requirements, and proposes a multi-time scale scheduling method and wind power prediction technology. In the short-term prediction technology, a day-to-day plan has been established to optimize the unit output of thermal generators, minimize operating costs and maximize the capacity to accommodate wind energy. Establish a time-of-day plan to optimize the power generation plans of all thermal, hydro and wind power plants on the basis of ultra-short-term wind power forecasting technology. However, in these studies, the attribute differences and energy dispatching characteristics of different energy sources are ignored, for example, the regulation of electric energy is more sensitive and its control mode is more flexible [22,23]; Thermal energy shows natural flexibility in regulation, with long response time and slow dynamic process [24]; However, natural gas needs to be added with conversion links during its use. On the premise that natural gas can be used in large quantities, it is inevitable to add large-capacity energy storage devices [25]; the cold energy change process is slow and requires a long dynamic time, which is not conducive to long-term storage [26]. At the same time, due to the strong coupling relationship of integrated energy systems, various energy systems influence each other, especially the non-linear devices of multiple coupled systems force the system to change frequently in operating load rate under the demand response, resulting in system "source-load" power imbalance. Therefore, based on the feasible interval conditions of system load rate, this paper studies the multi-time scale optimal operation of the cold-heat-electricity integrated energy system with the economic operation and the lowest pollutant gas emissions as multi-objective functions.

In the following content arrangement, Chapter 2 describes the topology and working principle of the cold-thermal-electricity integrated energy system. Chapter 3 analyzes the feasible range of load rate of integrated energy system. In Chapter 4, a multi-time scale optimization model is established with economic operation and minimum pollutant gas emissions as objective functions. In Chapter 5, the cold/heat/electricity front load of Jianguo hotel in Zhangjiakou, China is selected for simulation analysis and compared with a single time scale system. Chapter 6 summarizes this article.

## 2. Topological Structure and Working Principle of Cold-Thermal-Electricity Integrated Energy System

The integrated energy system has various structures, different forms and complicated coupling relationships, there are various energy production and conversion devices in the system. In this paper, the topology structure adopted for the research on the optimal operation of the cold-thermal-electricity integrated energy system is shown in Figure 1.

The cold-thermal-electricity integrated energy system is micro-energy network level. The equipment in the system mainly includes: gas internal combustion engine (GE), flue gas absorption heat pump (AP), cylinder liner water heat exchanger (JW), absorption refrigerator (AC), electric boiler (EB) and electric refrigerator (EC), and two kinds of energy storage equipment including electricity storage (ST) and heat storage (HS) are added. At the same time, in order to make full use of local reliable solar energy resources, a photovoltaic generator set (PV) is added and connected to the power network to ensure the balance of power supply and demand in the system. The whole system takes a gas internal combustion engine and a flue gas absorption heat pump as the core, the gas internal combustion engine directly supplies part of the electric load by consuming natural gas and generating electric energy, and the high-temperature steam generated during working is converted into hot water to supply the heat load through a cylinder liner water heat exchanger; The flue gas generated during

the combustion of natural gas can be absorbed and utilized by most of the flue gas absorption heat pumps and converted into heat energy and cold energy for direct supply to users; The absorption refrigerator converts part of the heat energy on the heat bus into cold energy to supply the cold load for use; When the system needs more heat energy or cold energy, some of the heat energy and cold energy deficiency can be made up through the work of the electric boiler or electric refrigerator. The system is also connected with two energy storage devices of heat storage and electricity storage to ensure that the system has sufficient margin of electric/thermal power capacity and increase the stability of the system. In addition, the active connection of photovoltaic generator sets not only effectively utilizes solar energy resources, but also increases the environmental protection and economic benefits of the system. When the demand for electric energy is large, the system can interact with the power grid. At the same time, in order to reduce the construction cost and coordination cost of the information channel and physical channel between the system and the power grid, the system adopts the principle of "connecting to the grid without power output " to purchase electric energy from the power grid, so as to make up for the shortage of electric energy in the system and ensure the stable operation of the system.

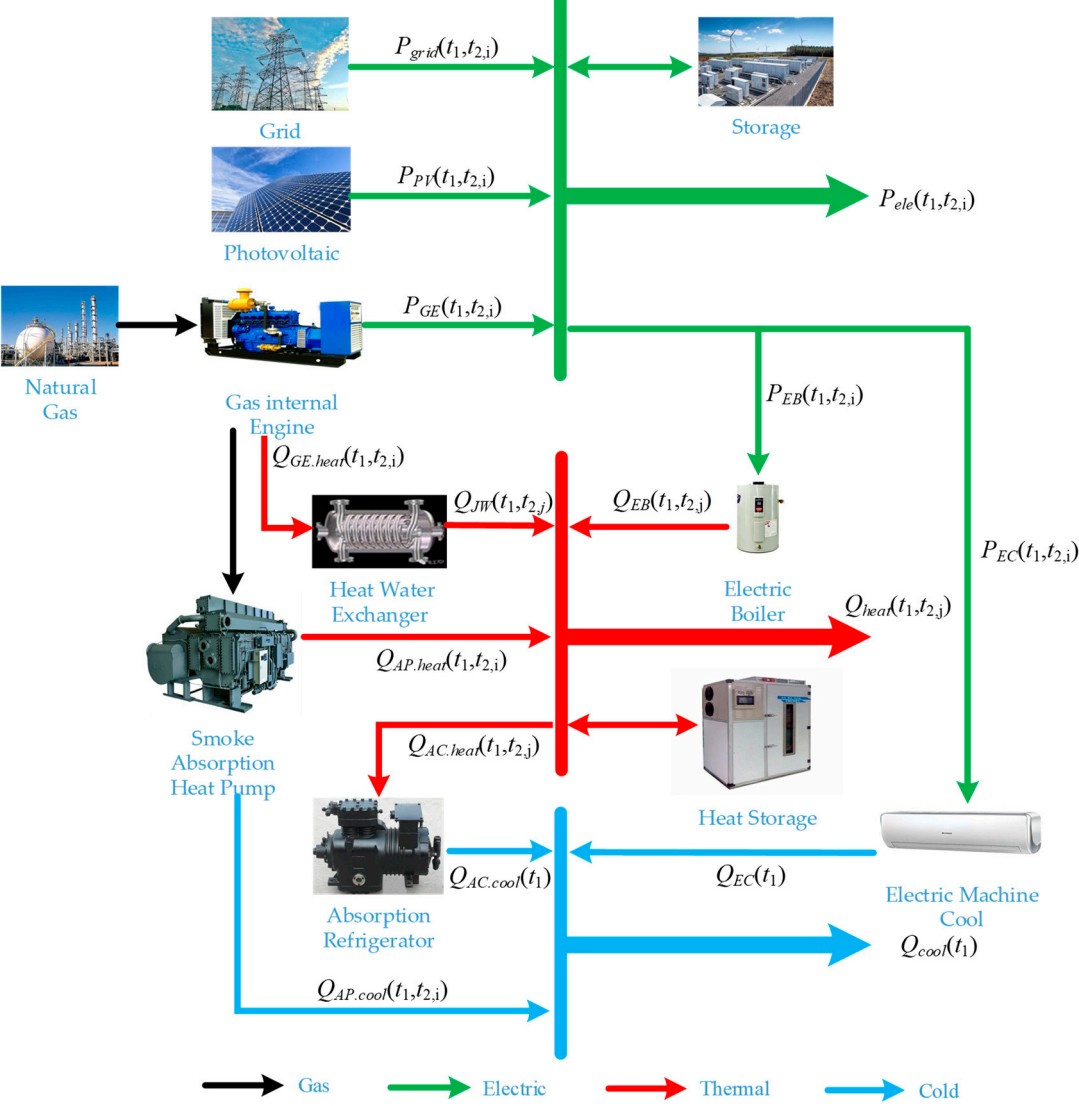

**Figure 1.** Topology diagram of cold-thermal-electricity integrated energy system.

## 3. The Feasible Range of System Load Rate

The integrated energy system realizes the coupling relation of multiple types of heterogeneous energy flows, and the system has strong coupling, nonlinearity, multi-energy complementation and mutual influence characteristics. For the cold-thermal-electricity integrated energy system, even if the operating characteristics and operating condition change parameters of each equipment in the system are clearly defined, there are many uncertainties in the energy output of the system under the condition that various energy input terminals are determined. Especially in the research process, it is found that when the cold-thermal-electricity integrated energy system is directly connected to the load side, the output power of the system does not match the load and the load rate of the system cannot find a reliable operation interval, so it cannot operate stably. Therefore, it is necessary to analyze the feasible range of system load rate.

Because of the close coupling relationship between the cold-thermal-electricity integrated energy system, it is difficult to directly analyze the load rate of the system. Therefore, it is possible to decouple the system and analyze the operational load rate characteristics of each decoupling subsystem, thus ensuring the "source-load" power balance of the system.

The decoupling method in this paper is as follows:

1. Build a system model according to the system topology, equipment and power constraints (see Chapter 4 for detailed models);
2. Change the system load factor A input from 0.1, 0.2 ... 1.0;
3. When the external power grid input is 0 KW, calculate the lowest lower limit of the cold, heat and electricity output power of each decoupling system;
4. When the external power grid input is 500 KW, calculate the maximum upper limit of the cooling, heating and electric output power of each decoupling system;
5. Power-load ratio curves of each decoupling subsystem are obtained.

Therefore, the feasible range of the load rate of each decoupling subsystem of the cold-thermal-electricity integrated energy system is shown in the following Figure 2:

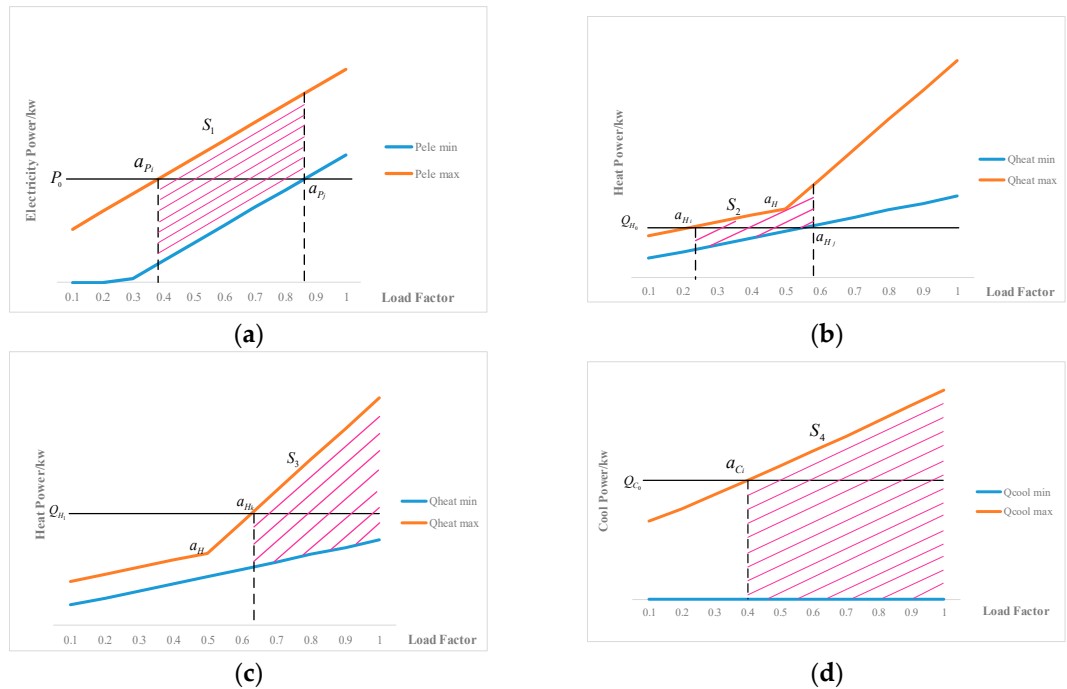

**Figure 2.** Load rate feasible interval of decoupling subsystem. (**a**) Decoupling power subsystem; (**b**) Decoupling the low load characteristics of the thermal energy subsystem; (**c**) Decoupling the high load characteristics of the thermal energy subsystem; (**d**) Decoupling the cold energy subsystem.

Since the loads in the cold-thermal-electricity integrated energy system are independent of each other and do not interfere with each other. For the analysis of the feasible range of system load rate, as shown in Figure 2a, when the electric power load is $P_0$, it intersects with the upper and lower limits of electric power at two points of load rate $a_{Pi}$ and $a_{Pj}$, then the feasible load rate of the power subsystem will be between ($a_{Pi}$, $a_{Pj}$), and the electric power output power is shown by shadow $S_1$ in the figure. There is a critical jump $Q_H$ in the upper limit of the power output of the decoupled thermal energy subsystem, where the system load rate is $a_H$. When the thermal energy load received by the system is lower than $Q_H$, it is called the low load characteristic of the thermal energy subsystem, and when the thermal energy load received by the system is higher than $Q_H$, it is called the high load characteristic of the thermal energy subsystem. As shown in Figure 2b, under the condition of low load characteristics, the upper and lower limits of the thermal energy load $Q_{H0}$ of the system and the power output of the thermal energy decoupling subsystem intersect at two points of the load rates $a_{Hi}$ and $a_{Hj}$, so that the system load rate is limited between ($a_{Hi}$, $a_{Hj}$), and the thermal energy output range of the thermal energy subsystem is shaded $S_2$ in Figure 2b; When the thermal energy load $Q_{H1}$ is higher than $Q_H$, the thermal energy subsystem exhibits a high load characteristic, as shown in Figure 2c, the load rate of the system will be higher than $a_{Hk}$, and the thermal energy output range is shown in shadow $S_3$. For the cold energy subsystem, as shown in Figure 2d, the lower limit of the cold energy output power is stable, while the upper limit of the output power varies approximately linearly, so that the cold energy system has a wider load rate selection range when outputting power outward. When the system cold energy load is $Q_{C0}$, the system load rate can fall between ($a_{Ci}$, 1), and the cold energy output is shown in shadow $S_4$. To sum up, the analysis of the operating characteristics of the load rate of each decoupling subsystem shows that the loads of inter-cold, heat and electricity in the system are independent and uncertain. In order to balance the "source-load" power of the system, the system load rate A should be in the same interval as the operational load rate of each decoupling subsystem, namely:

$$a \in (a_{Pi}, a_{Pj}) \cap (a_{Hi}, a_{Hj}) \cap (a_{Ci}, 1)$$
$$or \ \ a \in (a_{Pi}, a_{Pj}) \cap (a_{Hk}, \ 1) \cap (a_{Ci}, 1) \tag{1}$$

However, when the load rate intervals of each decoupling subsystem have no intersection, the "source-load" power of the system is unbalanced and cannot operate stably. Considering that the system load fluctuates within a stable range, it is possible to change the operating conditions of the system by adjusting the system equipment parameters and widen the intersection interval of load rates so that the system has a stable feasible interval. This paper will also study the optimal operation of the cold-thermal-electricity integrated energy system on the basis of the feasible load rate range of the system.

## 4. Optimize Operation

The operation process of the cold-thermal-electricity integrated energy system is complex, with many parameters and various energy outputs. This paper optimizes the operation of the cold-thermal-electricity integrated energy system with multi-objective and multi-time scales. In terms of multi-objective function, the system has the lowest operating cost and the least pollutant gas emission. In terms of multi-time scale, considering the difference of attributes of different energy sources and the different flexible characteristics of operation, for example, short power dispatching period, flexible regulation and fast power release and absorption, therefore, power is operated at $\Delta t_{2,i} = 15$ min as one dispatching time; However, heat energy is easier to adjust and store than cold energy. Therefore, heat energy is operated at $\Delta t_{2,j} = 30$ min as a scheduling time, while cold energy is operated at $\Delta t_1 = 1$ h as a scheduling time. In this way, the operation periods of cold, heat and electricity are multiple of each other, which is beneficial for multiple energy sources to supplement each other and ensures the consistency of multi-time scale coordination. At the same time, in order to reduce system losses and adjustment costs, the load rate of the system is kept unchanged for one hour, i.e., 4 periods of power adjustment.

*4.1. Objective Function*

The objective function is to construct a multi-objective function with the lowest total operating cost of the system and the lowest pollutant gas emissions, as follows:

$$\min \begin{cases} F_{run} = \sum_{t_1=1}^{24} \sum_{i=1}^{4} \left\{ F_{grid}(t_1, t_{2,i}) + F_{gas}(t_1, t_{2,i}) + F_{main}(t_1) \right\} \\ F_{poll} = \sum_{t_1=1}^{24} \sum_{i=1}^{4} \left\{ \alpha_{sour} \cdot P_{grid}(t_1, t_{2,i}) + \alpha_{tran} \cdot P_{grid}(t_1, t_{2,i}) + \alpha_{PGE} \cdot P_{GE}(t_1, t_{2,i}) \right\} \end{cases} \tag{2}$$

In the total operating cost, the electricity purchase cost, natural gas purchase cost and system equipment maintenance cost of the system and power grid are considered; Pollution gas emissions from the power supply side of the power grid, power loss from the power transmission lines of the power grid, and pollution gas emissions from the operation of the gas internal combustion engine are considered in the pollution gas emissions.

Where $F_{run}$ is the total operating cost of the system; $F_{grid}(t_1, t_{2,i})$ is the electricity purchase cost of the system and the power grid; $F_{gas}(t_1, t_{2,i})$ purchase natural gas for the system; $F_{main}(t_1)$ is the maintenance cost of system equipment; $F_{poll}$ is the amount of pollutant gas discharged; $\alpha_{sour}$ is the pollutant gas emission coefficient on the power supply side of the power grid; $\alpha_{tran}$ is the pollutant gas emission coefficient delivered by power grid lines; $P_{grid}(t_1, t_{2,i})$ purchases power for the system and power grid; $\alpha_{PGE}$ is the pollutant emission coefficient of gas internal combustion engine. $P_{GE}(t_1, t_{2,i})$ is the output electric power of the gas internal combustion engine; $t_1$ is expressed as the number of hours in a day. $t_2$ is expressed as the number of minutes in an hour. Since electric energy runs at $\Delta t_{2,i} = 15$ min as a scheduling time, electric energy is scheduled 4 times in an hour, $i = 1, 2, 3$ and 4. Similarly, heat energy is dispatched twice in one hour, with $j = 1$ and 2.

Among them, the system electricity purchase cost is specifically expressed as follows:

$$F_{grid}(t_1, t_{2,i}) = P_{grid}(t_1, t_{2,i}) \cdot \Delta t_{2,i} \cdot f_{grid}(t_1, t_{2,i}) \tag{3}$$

where, $f_{grid}(t_1, t_{2,i})$ is the real-time electricity price of the power grid.

The system purchase cost of natural gas is specifically expressed as follows:

$$F_{gas}(t_1, t_{2,i}) = V_{gas}(t_1, t_{2,i}) \cdot \Delta t_{2,i} \cdot f_{gas}(t_1, t_{2,i}) \tag{4}$$

where $V_{gas}(t_1, t_{2,i})$ is the volume of natural gas consumed by the system; $f_{gas}(t_1, t_{2,i})$ is the price of natural gas.

The system equipment maintenance costs are specified as follows:

$$F_{main}(t_1) = \sum_{i=1}^{4} \sum_{j=1}^{2} \left\{ \begin{array}{l} k_{GE}[P_{GE}(t_1, t_{2,i})] \cdot \Delta t_{2,i} \cdot P_{GE}(t_1, t_{2,i}) + k_{PV}[P_{PV}(t_1, t_{2,i})] \cdot \Delta t_{2,i} \cdot P_{PV}(t_1, t_{2,i}) + k_{AP.cool}[Q_{AP.cool}(t_1, t_{2,i})] \cdot \Delta t_{2,i} \cdot Q_{AP.cool}(t_1, t_{2,i}) + \\ k_{AP.heat}[Q_{AP.heat}(t_1, t_{2,i})] \cdot \Delta t_{2,i} \cdot Q_{AP.heat}(t_1, t_{2,i}) + k_{AC.heat}[Q_{AC.heat}(t_1, t_{2,j})] \cdot \Delta t_{2,j} \cdot Q_{AC.heat}(t_1, t_{2,j}) + \\ k_{batt.dis/cha}[P_{batt.dis/cha}(t_1, t_{2,i})] \cdot \Delta t_{2,i} \cdot P_{batt.dis/cha}(t_1, t_{2,i}) + k_{stor.dis/cha}[Q_{stor.dis/cha}(t_1, t_{2,j})] \cdot \Delta t_{2,j} \cdot Q_{stor.dis/cha}(t_1, t_{2,j}) \end{array} \right\} \tag{5}$$

where $k_{GE}[P_{GE}(t_1, t_{2,i})]$ is the maintenance coefficient of the gas internal combustion engine at different output powers; $k_{PV}[P_{PV}(t_1, t_{2,i})]$ is the maintenance coefficient of photovoltaic generator set; $P_{PV}(t_1, t_{2,i})$ is the generating power of photovoltaic generator set; $k_{AP.cool}[Q_{AP.cool}(t_1, t_{2,i})]$ is the cold power maintenance coefficient of the flue gas absorption heat pump; $Q_{AP.heat}(t_1, t_{2,i})$ is the cold power output by the flue gas absorption heat pump; $k_{AC.heat}[Q_{AC.heat}(t_1, t_{2,j})]$ is the thermal power maintenance coefficient of the flue gas absorption heat pump; $Q_{AP.heat}(t_1, t_{2,i})$ is the output heat power of the flue gas absorption heat pump; $k_{AC.heat}[Q_{AC.heat}(t_1, t_{2,j})]$ is the maintenance coefficient of absorption chiller; $Q_{AC.heat}(t_1, t_{2,j})$ is the heat power absorbed by the absorption refrigerator; $k_{batt.dis/cha}[P_{batt.dis/cha}(t_1, t_{2,i})]$, $k_{stor.dis/cha}[Q_{stor.dis/cha}(t_1, t_{2,j})]$ are the power maintenance coefficients of power storage equipment and heat storage equipment respectively; $P_{batt.dis/cha}(t_1, t_{2,i})$ and $Q_{stor.dis/cha}(t_1, t_{2,j})$ are the interactive power of electric storage equipment and heat storage equipment.

*4.2. Constraints*

The cold-thermal-electric integrated energy system constraints include: equipment model constraints and power balance constraints.

4.2.1. Equipment Model Constraints

Among them, the equipment model includes a gas internal combustion engine model [27,28], a flue gas absorption heat pump model [29–31], an absorption refrigerator model [32], a cylinder liner water heat exchanger model, an electric boiler model, an electric refrigerator model, a photovoltaic generator set model [33], a heat and electricity storage model [34–36].

- Gas engine model.

Among them, gas-fired internal combustion engines have good electrical energy and thermal energy output characteristics which can be divided into power generation, heat generation and consumption of natural gas.

Power generation of gas internal combustion engine, the maximum real-time output power of an internal combustion engine is limited to $P_{\max}$:

$$
\begin{cases}
\eta_{GE.elec}(t_1, t_{2,i}) = a_3 \left( \frac{P_{GE}(t_1,t_{2,i})}{P_{\max}} \right)^3 + a_2 \left( \frac{P_{GE}(t_1,t_{2,i})}{P_{\max}} \right)^2 + a_1 \left( \frac{P_{GE}(t_1,t_{2,i})}{P_{\max}} \right) + a_0 \\
P_{GE}(t_1, t_{2,1}) = P_{GE}(t_1, t_{2,2}) = P_{GE}(t_1, t_{2,3}) = P_{GE}(t_1, t_{2,4}) \\
\left| P_{GE}(t_1, t_{2,i}) - P_{GE}(t_1, t_{2,i} - 1) \right| \leq P_{GE.\max}
\end{cases}
\tag{6}
$$

Heating power of gas internal combustion engine:

$$
\begin{cases}
Q_{GE.heat}(t_1, t_{2,i}) = \frac{P_{GE}(t_1,t_{2,i})}{\eta_{GE.elec}(t_1,t_{2,i})} \left( 1 - \eta_{GE.elec}(t_1, t_{2,i}) - \eta_L \right) \\
Q_{GE.heat}(t_1, t_{2,1}) = Q_{GE.heat}(t_1, t_{2,2}) = Q_{GE.heat}(t_1, t_{2,3}) = Q_{GE.heat}(t_1, t_{2,4})
\end{cases}
\tag{7}
$$

Natural gas consumed by gas internal combustion engines:

$$
V_{gas}(t_1, t_{2,i}) = \frac{P_{GE}(t_1, t_{2,i}) \cdot \Delta t_2}{\eta_{gas} \cdot \eta_{GE.elec}(t_1, t_{2,i}) \cdot LHV}
\tag{8}
$$

In the formula, the output electric power $P_{GE}(t_1,t_{2,i})$ of the gas internal combustion engine remains unchanged for one hour, i.e., four periods of electric energy output; $\eta_{GE.elec}(t_1,t_{2,i})$ is the power generation efficiency of the gas internal combustion engine; $P_{\max}$ is the rated power of gas internal combustion engine. The thermal power $Q_{GE.heat}(t_1,t_{2,i})$ output by the gas internal combustion engine remains constant within one hour. $\eta_L$ is the inherent loss rate of gas internal combustion engine; $P_{GE.\max}$ is the output gradient constraint of gas internal combustion engine. LHV is the low calorific value of natural gas; $\eta_{gas}$ is the natural gas utilization rate of gas internal combustion engines; $a_3$, $a_2$, $a_1$ and $a_0$ are fitting constants respectively.

- Smoke absorption heat pump model.

Smoke absorption heat pump model can be divided into three parts: absorbing smoke, outputting heat energy and cold energy.

Smoke absorption heat pump absorbs smoke, absorption heat pump absorbed the flue gas temperature are directly affected parameter $COP_{AP}$:

$$
\begin{cases}
T(t_1, t_{2,i}) = b_1 \cdot \left( \frac{P_{GE}(t_1,t_{2,i})}{P_{\max}} \right) + b_0 \\
C_w(t_1, t_{2,i}) = b_3 \cdot T(t_1, t_{2,i}) + b_2 \\
COP_{AP}(t_1, t_{2,i}) = b_5 \cdot \left( \frac{P_{GE}(t_1,t_{2,i})}{P_{\max}} \right)^{b_4} \\
\lambda_{heat}(t_1, t_{2,i}) + \lambda_{cool}(t_1, t_{2,i}) = 1
\end{cases}
\tag{9}
$$

At the same time, the real-time power of flue gas absorption heat pump has linear constraint values $Q_{AP.heat.\max}$ and $Q_{AP.cool.\max}$.

Smoke absorbs heat output from heat pump:

$$\begin{cases} Q_{AP.heat}(t_1,t_{2,i}) = \lambda_{heat}(t_1,t_{2,i}) \cdot C_w(t_1,t_{2,i}) \cdot (T(t_1,t_{2,i}) - T_{heat}) \cdot COP_{AP}(t_1,t_{2,i}) \cdot L_{heat}(t_1,t_{2,i}) \cdot \eta_{AP.heat} \\ Q_{AP.heat}(t_1,t_{2,1}) = Q_{AP.heat}(t_1,t_{2,2}) = Q_{AP.heat}(t_1,t_{2,3}) = Q_{AP.heat}(t_1,t_{2,4}) \\ \left| Q_{AP.heat}(t_1,t_{2,i}) - Q_{AP.heat}(t_1,t_{2,i}-1) \right| \leq Q_{AP.heat.\max} \\ 0 \leq L_{heat}(t_1,t_{2,i}) \leq L_{heat.\max} \end{cases} \quad (10)$$

Smoke absorption heat pump outputs cold energy:

$$\begin{cases} Q_{AP.cool}(t_1,t_{2,i}) = \lambda_{cool}(t_1,t_{2,i}) \cdot C_w(t_1,t_{2,i}) \cdot (T(t_1,t_{2,i}) - T_{cool}) \cdot COP_{AP}(t_1,t_{2,i}) \cdot L_{cool}(t_1,t_{2,i}) \cdot \eta_{AP.cool} \\ Q_{AP.cool}(t_1,t_{2,1}) = Q_{AP.cool}(t_1,t_{2,2}) = Q_{AP.cool}(t_1,t_{2,3}) = Q_{AP.cool}(t_1,t_{2,4}) \\ \left| Q_{AP.cool}(t_1,t_{2,i}) - Q_{AP.cool}(t_1,t_{2,i}-1) \right| \leq Q_{AP.cool.\max} \\ 0 \leq L_{cool}(t_1,t_{2,i}) \leq L_{cool.\max} \end{cases} \quad (11)$$

where in $T(t_1,t_{2,i})$ is the inlet temperature of the flue gas absorption heat pump; $C_W(t_1,t_{2,i})$ is the specific heat capacity of hot water at different temperatures; $COP_{AP}(t_1,t_{2,i})$ is the energy efficiency coefficient of flue gas absorption heat pump; The heating power $Q_{AP.heat}(t_1,t_{2,i})$ and the cooling power $Q_{AP.cool}(t_1,t_{2,i})$ of the flue gas absorption heat pump remain unchanged within one hour. $\lambda_{heat}(t_1,t_{2,i})$ and $\lambda_{cool}(t_1,t_{2,i})$ are the heating ratio and cooling ratio of flue gas of the flue gas absorption heat pump respectively; $T_{heat}$ and $T_{cool}$ are hot water outlet temperature and cold water outlet temperature respectively. $L_{heat}(t_1,t_{2,i})$ and $L_{cool}(t_1,t_{2,i})$ are the hot water and cold water flows of the flue gas absorption heat pump respectively; $L_{heat.\max}$ and $L_{cool.\max}$ are the maximum heating and cooling flows respectively; $\eta_{AP.heat}$ and $\eta_{AP.cool}$ are the heating and cooling efficiency of flue gas absorption heat pump respectively. $Q_{AP.heat.\max}$ is the gradient constraint of heating power output of flue gas absorption heat pump; $Q_{AP.cool.\max}$ is the gradient constraint of cooling power output of flue gas absorption heat pump; $b_5$, $b_4$, $b_3$, $b_2$, $b_1$ and $b_0$ are fitting constants respectively.

- Absorption refrigerator model.

$$\begin{cases} Q_{AC.cool}(t_1) = COP_{AC} \cdot \sum_{j=1}^{2} Q_{AC.heat}(t_1,t_{2,j}) \\ Q_{AC.heat.\min} \leq Q_{AC.heat}(t_1,t_{2,j}) \leq Q_{AC.heat.\max} \\ \left| Q_{AC.cool}(t_1) - Q_{AC.cool}(t_1-1) \right| \leq Q_{AC.cool.\max} \end{cases} \quad (12)$$

where $Q_{AC.cool}(t_1)$ is the cold power output by the absorption refrigerator; $COP_{AC}$ is the energy efficiency coefficient of absorption refrigerator; $Q_{AC.heat.\min}$ and $Q_{AC.heat.\max}$ are the minimum and maximum thermal power absorbed by the absorption chiller respectively. $Q_{AC.cool.\max}$ is the output gradient constraint of absorption chiller.

In order to stabilize the output power of absorption refrigerator, the absolute value of the difference between the next output power and the previous one is $Q_{AC.cool.\max}$.

- Cylinder liner water heat exchanger model.

$$Q_{JW}(t_1,t_{2,j}) = \eta_{JW} \cdot \sum_{i=2j-1}^{2j} Q_{GE}(t_1,t_{2,i}) \quad (13)$$

where $Q_{JW}(t_1,t_{2,j})$ is the output thermal power of the cylinder liner water heat exchanger; $\eta_{JW}$ is the heat transfer efficiency of cylinder liner water heat exchanger.

- Electric boiler model.

$$\begin{cases} Q_{EB}(t_1, t_{2,j}) = COP_{EB} \cdot \sum_{i=2j-1}^{2j} P_{EB}(t_1, t_{2,i}) \\ P_{EB.\min} \le P_{EB}(t_1, t_{2,i}) \le P_{EB.\max} \\ \left| Q_{EB}(t_1, t_{2,j}) - Q_{EB}(t_1, t_{2,j} - 1) \right| \le Q_{EB.\max} \end{cases} \tag{14}$$

In the formula, $P_{EB}(t_1, t_{2,i})$ is the input electric power of the electric boiler; $Q_{EB}(t_1, t_{2,j})$ is the output thermal power of the electric boiler; $COP_{EB}$ is the energy production coefficient of electric boilers; $P_{EB.\min}$ and $P_{EB.\max}$ are respectively the minimum and maximum electric power of electric boilers; $Q_{EB.\max}$ is the output gradient constraint of the electric boiler.

- Electric refrigerator model.

$$\begin{cases} Q_{EC}(t_1) = COP_{EC} \cdot \sum_{i=1}^{4} P_{EC}(t_1, t_{2,i}) \\ P_{EC.\min} \le P_{EC}(t_1, t_{2,i}) \le P_{EC.\max} \\ \left| Q_{EC}(t_1) - Q_{EC}(t_1 - 1) \right| \le Q_{EC.\max} \end{cases} \tag{15}$$

where $P_{EC}(t_1, t_{2,i})$ is the input electric power of the electric refrigerator; $Q_{EC}(t_1)$ is the output cold power of the electric refrigerator; $COP_{EC}$ is the energy efficiency coefficient of electric refrigerator. $P_{EC.\min}$ and $P_{EC.\max}$ are the minimum and maximum electric power of electric refrigerator respectively. $Q_{EC.\max}$ is the output gradient constraint of the electric refrigerator.

- Photovoltaic generator set model.

$$P_{PV}(t_1, t_{2,i}) = P_{STC} \frac{G_{ING}(t_1, t_{2,i})}{G_{STC}} [1 - k(T_{out}(t_1, t_{2,i}) - T_s)] \tag{16}$$

where, $P_{STC}$ is the rated output of photovoltaic generator set; $G_{ING}(t_1, t_{2,i})$ is the real-time irradiation intensity; $G_{STC}$ is the rated irradiation intensity of photovoltaic generator set; $K$ is the power generation coefficient of photovoltaic generator set; $T_{out}(t_1, t_{2,i})$ is the external temperature; $T_s$ is the reference temperature of the generator set.

- Electricity storage model.

In the model of electricity storage model, the power balance and charge and discharge constraints are considered.

$$\begin{cases} E_{batt}(t_1, t_{2,i}) = (1 - k_L) \cdot E_{batt}(t_1, t_{2,i} - 1) + [\eta_{batt.cha} \cdot P_{batt.cha}(t_1, t_{2,i}) - \frac{P_{batt.dis}(t_1, t_{2,i})}{\eta_{batt.dis}}] \cdot \Delta t_{2,i} \\ P_{batt.cha.\min} \le P_{batt.cha}(t_1, t_{2,i}) \le P_{batt.cha.\max} \\ P_{batt.dis.\min} \le P_{batt.dis}(t_1, t_{2,i}) \le P_{batt.dis.\max} \\ E_{batt.\min} \le E_{batt}(t_1, t_{2,i}) \le E_{batt.\max} \end{cases} \tag{17}$$

where in $E_{batt}(t_1, t_{2,i})$ is that real-time capacity of the pow storage equipment; $k_L$ is the self-loss coefficient of power storage equipment; $\eta_{batt.cha}$ and $\eta_{batt.dis}$ are the charging and discharging efficiency of power storage equipment respectively. $P_{batt.cha}(t_1, t_{2,i})$ and $P_{batt.dis}(t_1, t_{2,i})$ are the charging and discharging power of the power storage equipment respectively. $P_{batt.dis.\max}$ and $P_{batt.dis.\min}$ are the maximum and minimum discharge powers of power storage equipment respectively. $P_{batt.cha.\max}$ and $P_{batt.cha.\min}$ are respectively the maximum and minimum charging power of power storage equipment. $E_{batt.\max}$ and $E_{batt.\min}$ are the maximum and minimum storage capacities of power storage equipment respectively.

- Thermal storage model.

  Thermal storage model have similar model constraints to electricity storage model.

$$
\begin{cases}
B_{stor}(t_1, t_{2,j}) = (1 - k_s) \cdot B_{stor}(t_1, t_{2,j} - 1) + \left[ \eta_{stor.cha} \cdot Q_{stor.cha}(t_1, t_{2,j}) - \dfrac{Q_{stor.dis}(t_1, t_{2,j})}{\eta_{stor.dis}} \right] \cdot \Delta t_{2,j} \\
Q_{stor.cha.\min} \leq Q_{stor.cha}(t_1, t_{2,j}) \leq Q_{stor.cha.\max} \\
Q_{stor.dis.\min} \leq Q_{stor.dis}(t_1, t_{2,j}) \leq Q_{stor.dis.\max} \\
B_{stor.\min} \leq B_{stor}(t_1, t_{2,j}) \leq B_{stor.\max}
\end{cases}
\tag{18}
$$

where $B_{stor}(t_1, t_{2,j})$ is the real-time capacity of heat storage equipment; $k_s$ is the self-loss coefficient of heat storage equipment; $\eta_{stor.cha}$ and $\eta_{stor.dis}$ are the heat absorption and heat release efficiency of heat storage equipment respectively. $Q_{stor.cha}(t_1, t_{2,j})$ and $Q_{stor.dis}(t_1, t_{2,j})$ are the heat absorption and heat release power of heat storage equipment respectively. $Q_{stor.cha.\max}$ and $Q_{stor.cha.\min}$ are the maximum and minimum heat absorption powers of heat storage equipment respectively. $Q_{stor.dis.\max}$ and $Q_{stor.dis.\min}$ are the maximum and minimum heat release power of heat storage equipment respectively. $B_{stor.\max}$ and $B_{stor.\min}$ are the maximum and minimum capacity constraints of heat storage equipment respectively.

### 4.2.2. Power Balance Constraints

The power balance constraints of cold, heat and electricity are satisfied in the system.

- Electric power balance constraint.

$$
\begin{aligned}
P_{grid}(t_1, t_{2,i}) + P_{PV}(t_1, t_{2,i}) \quad &+ P_{GE}(t_1, t_{2,i}) + P_{batt.dis}(t_1, t_{2,i}) \cdot D_{batt.dis}(t_1, t_{2,i}) = \\
&P_{batt.cha}(t_1, t_{2,i}) \cdot D_{batt.cha}(t_1, t_{2,i}) + P_{ele}(t_1, t_{2,i}) + P_{EB}(t_1, t_{2,i}) + P_{EC}(t_1, t_{2,i})
\end{aligned}
\tag{19}
$$

In the formula, $D_{batt.dis}(t_1, t_{2,i})$ and $D_{batt.cha}(t_1, t_{2,i})$ are respectively discharge and charge variables of power storage equipment; $P_{ele}(t_1, t_{2,i})$ is the power load.

- Thermal power balance constraint.

$$
\begin{aligned}
Q_{JW}(t_1, t_{2,j}) + \sum_{i=2j-1}^{2j} Q_{AP.heat}(t_1, t_{2,i}) \quad &+ Q_{EB}(t_1, t_{2,j}) + Q_{stor.dis}(t_1, t_{2,j}) \cdot D_{stor.dis}(t_1, t_{2,j}) = \\
&Q_{stor.cha}(t_1, t_{2,j}) \cdot D_{stor.cha}(t_1, t_{2,j}) + Q_{heat}(t_1, t_{2,j}) + Q_{AC.heat}(t_1, t_{2,j})
\end{aligned}
\tag{20}
$$

In the formula, $D_{stor.dis}(t_1, t_{2,j})$ and $D_{stor.cha}(t_1, t_{2,j})$ are the heat release and heat absorption variables of the heat storage equipment respectively; $Q_{heat}(t_1, t_{2,j})$ is thermal load.

- Cold power balance constraint.

$$
Q_{AC.cool}(t_1) + Q_{EC}(t_1) + \sum_{i=1}^{4} [Q_{AP.cool}(t_1, t_{2,i})] = Q_{cool}(t_1)
\tag{21}
$$

where $Q_{cool}(t_1)$ is the cooling load.

### 4.3. Solution

#### 4.3.1. Multi-Objective Solution Method

This model is a multi-objective mixed integer nonlinear programming model. The multi-objective problem is transformed into a single-objective problem that is easy to solve by adopting a scalar linear weighting method. The results under different weight conditions are compared by changing the weight coefficient to obtain the optimal result. The scalar process is as follows:

Firstly, the optimal values of economic operation and pollutant gas emission under single-objective conditions are solved respectively, and $F_{run.min}$ and $F_{poll.min}$ are obtained.

$$\min \; F_{run} = \sum_{t_1=1}^{24} \sum_{i=1}^{4} \left\{ F_{grid}(t_1, t_{2,i}) + F_{gas}(t_1, t_{2,i}) + F_{main}(t_1) \right\} \tag{22}$$

$$\min \; F_{poll} = \sum_{t_1=1}^{24} \sum_{i=1}^{4} \left\{ \alpha_{sour} \cdot P_{grid}(t_1, t_{2,i}) + \alpha_{tran} \cdot P_{grid}(t_1, t_{2,i}) + \alpha_{PGE} \cdot P_{GE}(t_1, t_{2,i}) \right\} \tag{23}$$

Secondly, the multi-objective optimization problem is transformed into a single-objective problem solving calculation through a linear weighting method. The solving process is as follows:

$$\min F = k_{run} \cdot \frac{F_{run}}{F_{run.min}} + k_{poll} \cdot \frac{F_{poll}}{F_{poll.min}} , \; k_{run} + k_{poll} = 1 \tag{24}$$

where $F$ is the mixed objective function value; $k_{run}$ is the economic operation weight coefficient; $k_{poll}$ is the weight coefficient of pollutant gas emission.

By changing the values of the weight coefficients $k_{run}$ and $k_{poll}$ to adjustment and optimization results of the objective function are compared, the sum of weight coefficients is always 1. In this paper, 0.1 is selected as the discrimination degree of two parameters. Therefore, the value of the weight coefficient $(k_{run}, k_{poll})$ = (0.1,0.9), (0.2,0.8) … (0.9,0.1) changes.

Finally, by changing different weight coefficients, the optimal operating conditions under different weights are calculated, and the result analysis is obtained.

### 4.3.2. Model Optimization Process

This model is programmed by LINGO software and solved by GLOBAL algorithm. The model optimization process is shown in Figure 3 below.

First of all, initialization the system, system variables and parameters are defined, input cold, thermal, electric load parameters and $k_{run}/k_{poll}$ weight parameters, then, objective function, equipment model constraint and power balance constraint are input.

After that, the LINGO software is used to calculate the optimization model. If there is no feasible solution to the optimization model, then there is no solution to the original optimization problem; otherwise, continue to optimize the process. The results of feasible solutions are compared. If the result of comparison is the minimum of feasible solutions, the global optimal solution is obtained. Instead, continue to compare possible solutions.

Finally, output the optimal result and end the optimization process.

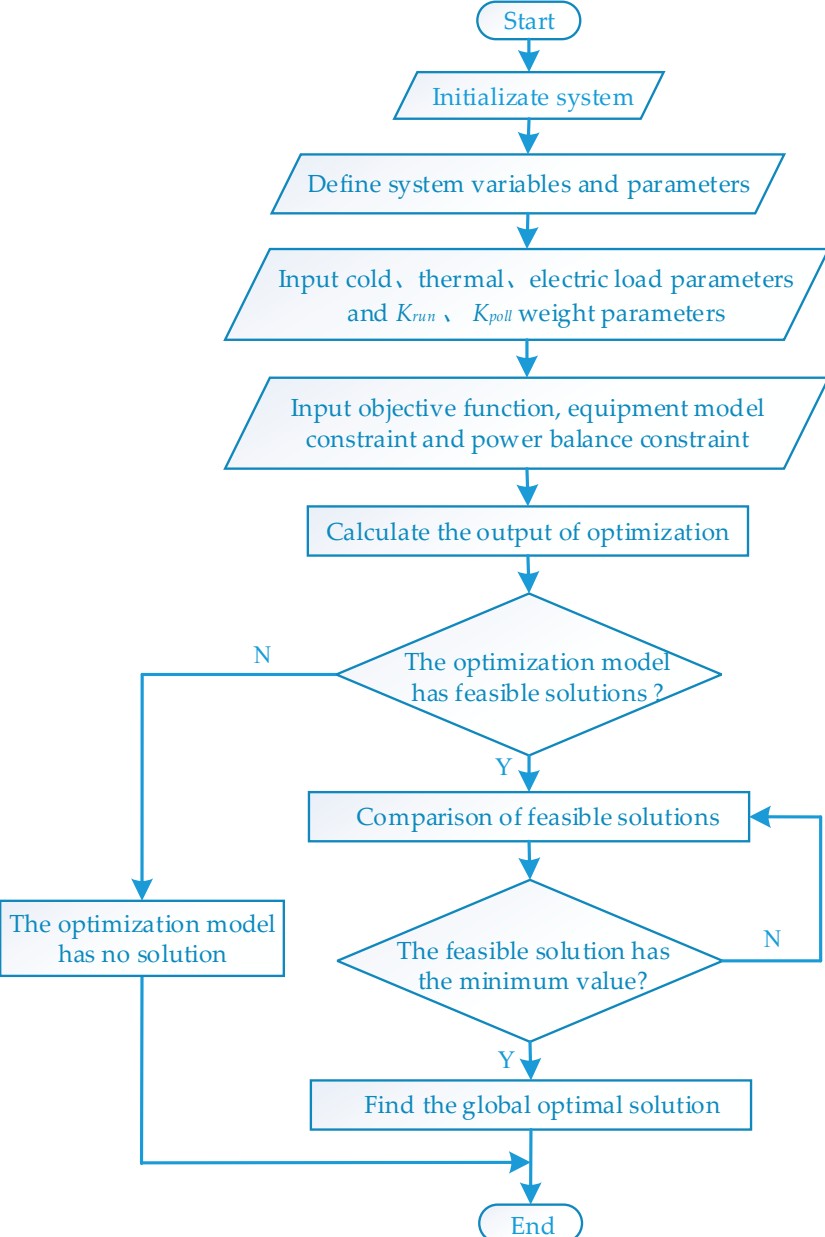

**Figure 3.** Model optimization process.

## 5. Numerical Simulation and Operation Load Rate Analysis

### 5.1. Numerical Simulation and Operation Results Analysis

This paper selects Jianguo hotel in Zhangjiakou, northern China, to analyze the system's cold, heat and electricity load before summer. At the same time, considering the feasible range of system load rate, the system operating conditions are adjusted by changing the system equipment parameters, thus expanding the system's operational load rate and balancing the system's "source-load" power.

The parameters of electricity, heat and cold load in summer are shown in the following Figures 4–6:

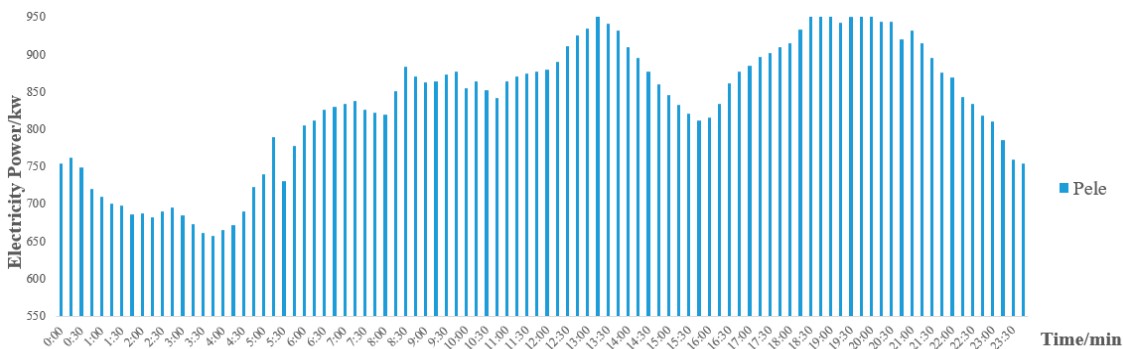

**Figure 4.** Summer electricity power load chart.

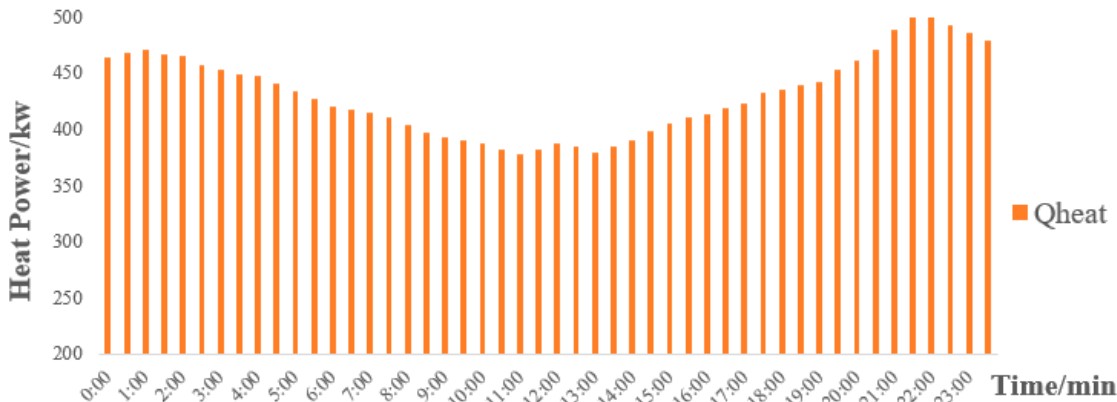

**Figure 5.** Summer heat load chart.

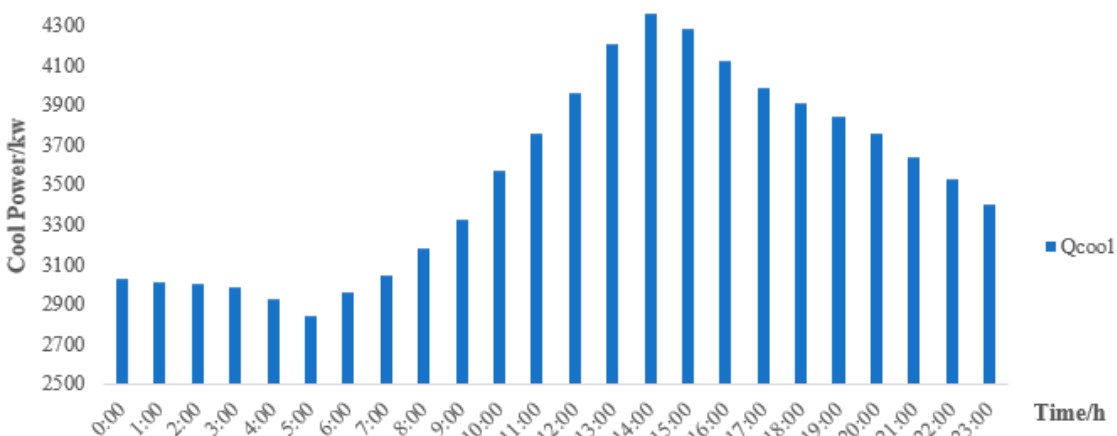

**Figure 6.** Summer cooling load chart.

The electricity power load before summer presents obvious "peak and valley" characteristics. The power demand is large during peak hours and low during valley hours. Therefore, it is very suitable for implementing time-of-use electricity price. The specific time-of-use electricity price is shown in Table 1 below.

**Table 1.** Time-of-use tariff.

| Name | Time/h | Price/CNY |
| --- | --- | --- |
| peak period | 8~10, 18~19 | 0.866 |
| peacetime period | 11~17, 20~22 | 0.559 |
| valley period | 0~7, 23 | 0.223 |

In summer, the system has a low demand for heat energy, and the load fluctuates. The temperature rises from 10: 00 to 14: 00 noon, and the heat energy load is low. The system has a high demand for hot water and a high heat energy load from 21: 00 to 23: 00 at night.

In summer, the system has a strong demand for cool energy. In the early morning (0:00~5:00), the cool energy load is relatively low, then increases with the temperature rising, reaches the maximum at 14:00, and then continues to decline.

This optimization problem is a mixed integer nonlinear problem. Standard linear method is adopted to solve the multi-objective problem. By changing the weight coefficients of $k_{run}$ and $k_{poll}$, different optimization results are compared, and the global optimal solution is obtained. In terms of optimization model calculation, this model has a total of 1923 variables and 1824 constraints, including 1128 linear variables and 766 linear constraints. In order to ensure that the output result is the global optimal solution, multiple feasible solutions need to be compared, so the calculation time is also relatively long. The calculation time for each group of different weight coefficients needs at least 3–4 h. The optimal results are obtained by changing the $k_{run}$ and $k_{poll}$ weight coefficients of multi-objective functions as shown in Figure 7 below:

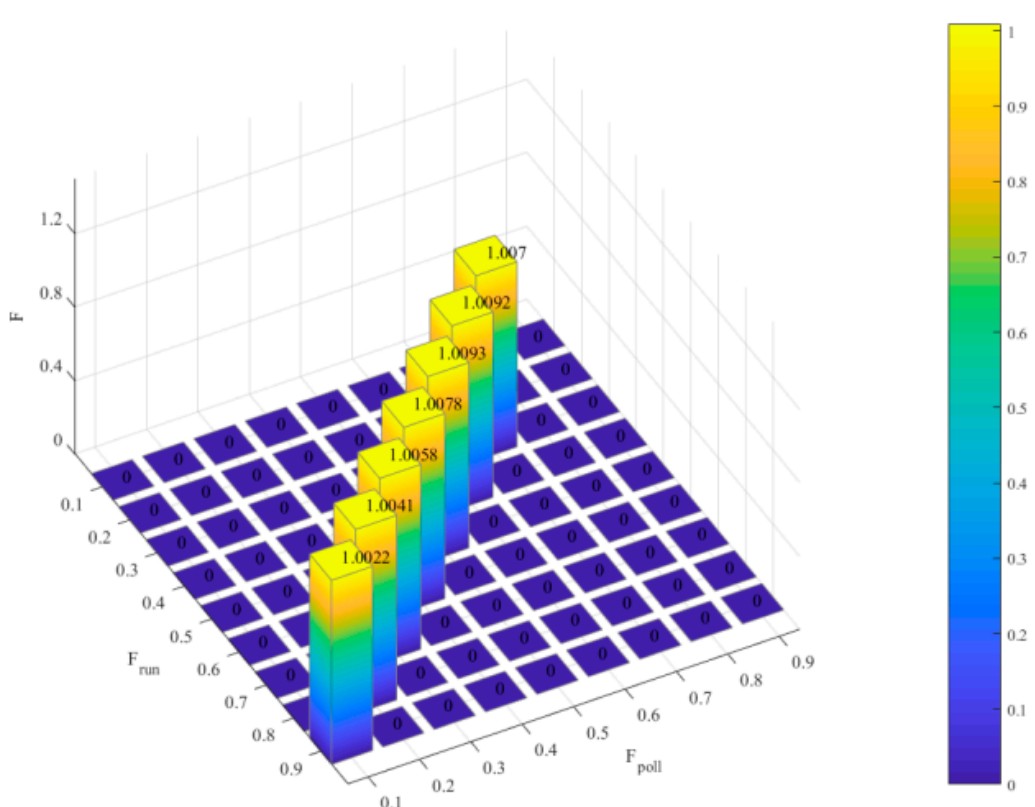

**Figure 7.** Optimal operation results of different weights in summer.

According to the optimization results of different weights in summer in Figure 7, when the weights $(k_{run}, k_{poll})$ = (0.1,0.9) and (0.2,0.8), the system cannot operate, and the weights of these two points should be discarded. When the weights $k_{run}$ = 0.9 and $k_{poll}$ = 0.1, the mixed objective function value $F$ is the minimum of 1.0022. At this time, the electric/heat/cool output power of each equipment before summer is shown in the following Figures 8–10:

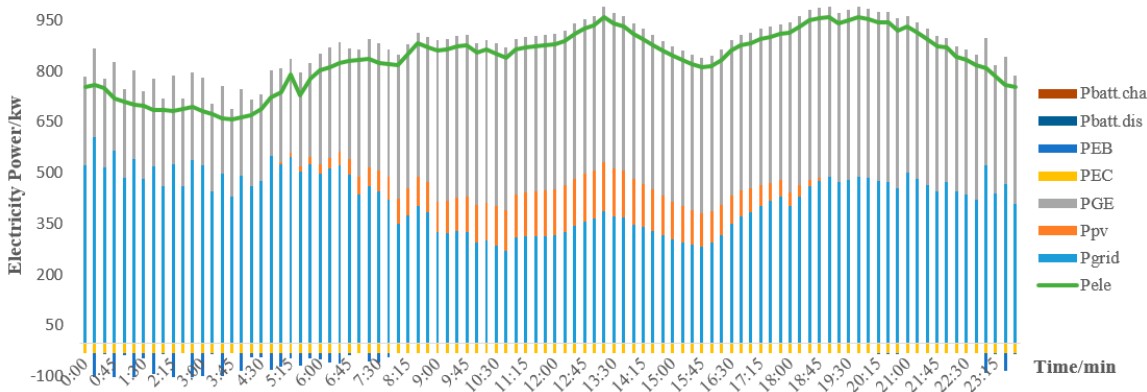

**Figure 8.** Power output diagram of various equipment before multi-time scale in summer.

As can be seen from Figure 8, the gas internal combustion engine has maintained good electric energy output characteristics. In the 0th to 7th and 23rd hour, the electricity price of the power grid is low, which is suitable for the system to receive a large amount of electric energy of the power grid and meets the requirements of system economy. The gas internal combustion engine is also put into operation, with the starting load rate kept above 55% and the output power stable. In the 8th to 17th hour, the photovoltaic unit is actively connected to the system, the system fully absorbs renewable energy, the power of the power grid is greatly reduced, and the gas internal combustion engine runs stably. At this time, the load rate is kept above 85%, and the electric energy output is stable. During the 18th to 22nd hour of the peak power consumption at night, although the photovoltaic unit will no longer output power, the gas internal combustion engine will remain in a state close to full load output and the power output of the power grid will be limited to a stable range. Electric boilers and electric refrigerators, as units of electric energy consumption, are converted into heat energy and cold energy to supply loads for use. At the same time, the charging and discharging power of the power storage equipment is less throughout the day, which indicates that the system can realize self-absorption, which is beneficial to prolonging the service life of the power storage equipment and the stable operation of the system power.

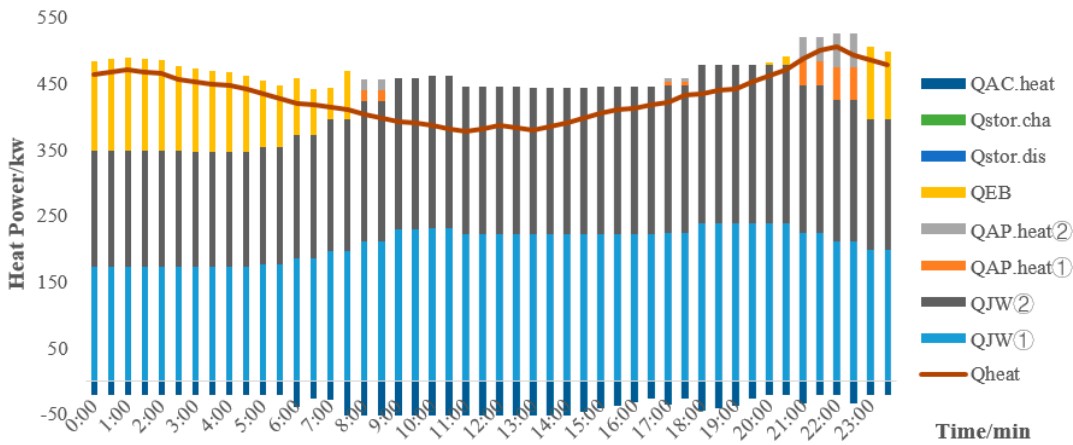

**Figure 9.** Heat output diagram of various equipment before multi-time scale in summer.

In summer, the demand for heat energy load is relatively low. The heat energy supplied by converting the cylinder liner water directly connected with the gas internal combustion engine into hot water basically bears the base load part of the heat energy load in summer. The electric energy operates twice in one heat energy operation cycle, while the gas internal combustion engine outputs stably in one hour. Therefore, the heat power output by the cylinder liner water is relatively stable in one heat

energy operation cycle. The electric boiler works in the electricity valley period and supplements the heat energy load with the flue gas absorption heat pump. The absorption refrigerator absorbs the surplus heat energy from the cylinder liner water during the summer heat energy valley and converts it into cold energy. The heat storage equipment absorbs and releases less heat energy during one day's operation, which indicates that the system has balanced heat power and stable operation.

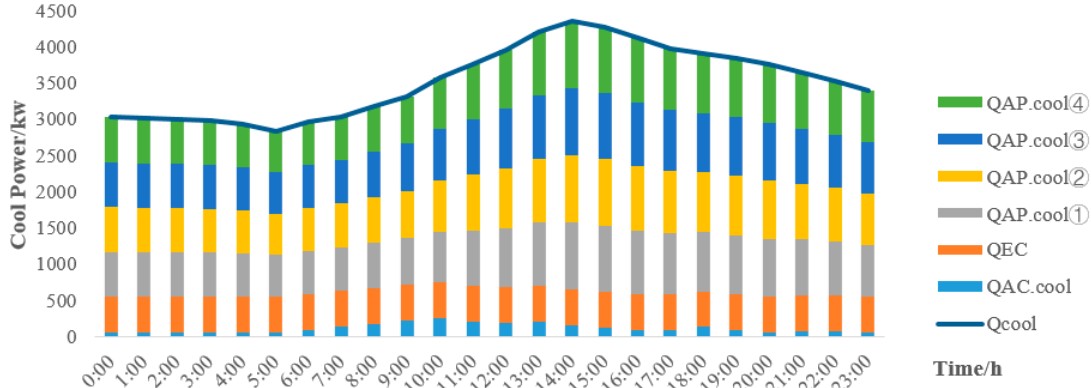

**Figure 10.** Cooling output diagram of various equipment before multi-time scale in summer.

In summer, the demand for cooling energy load is large, and the electric refrigerator and absorption refrigerator provide only a small part of the cooling energy load all day long. As the flue gas absorption heat pump directly absorbs the flue gas heat energy of the gas internal combustion engine, it can stably output four times of cooling energy in one cooling energy scheduling cycle. Moreover, the flue gas absorption heat pump is sensitive to load changes, has good load following characteristics, and meets the cooling power load demand of the system.

*5.2. Analysis of Operation Load Rate Results*

In terms of system operation load rate, the loads of cold, heat and electricity are independent of each other. Since the system load rate is affected by the loads of electricity, heat and cold at the same time, analyzing the changes of the three to the load rate will directly determine the stable operation of the system. The operation of electric/heat/cold load and load rate is shown in Figures 11–13.

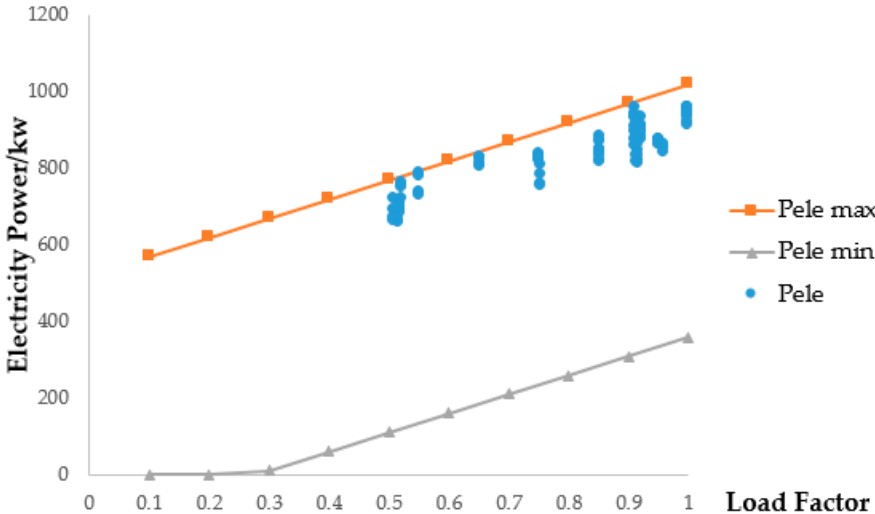

**Figure 11.** Multi-time scale power load-load rate operation.

As shown in Figure 11, the multi-time scale power load and load rate operating points are mostly close to the upper power limit of the decoupled power system, while the load rate operating points close to the upper power limit are close to full load operation, and the system is stable at this time. The closer the operating point is to the upper limit of power, the more thorough the system is running, the greater the utilization rate of power and the higher the economy of the system.

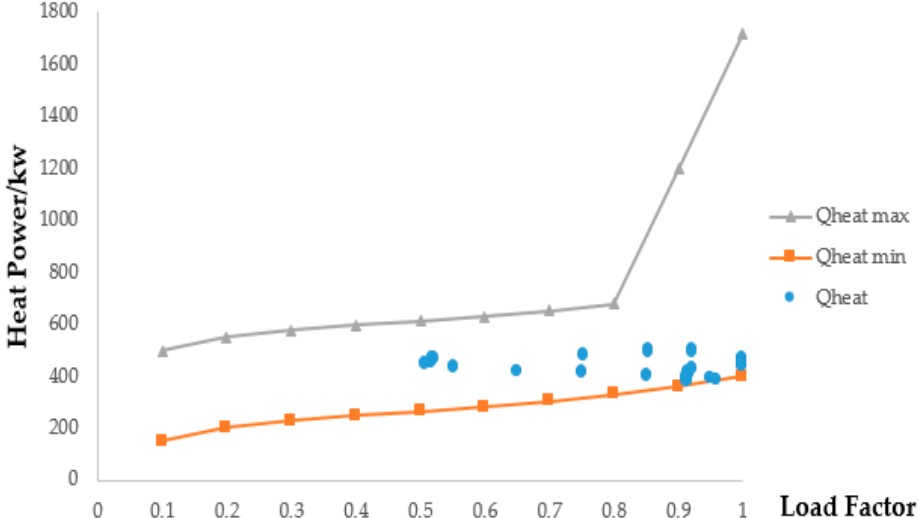

**Figure 12.** Multi-time scale heat load-load rate operation.

The multi-time scale heat load and load rate operation is shown in Figure 12. The summer heat load demand is low and the load fluctuation range is small, so the heat load and load rate operation points are distributed in a scatter line within the thermal energy output range. At the higher load rate, the operating point is close to the lower limit of the heat energy output boundary, but the heat storage equipment has not been put into operation at this time, indicating that the "source-charge" power of the heat energy subsystem is still balanced.

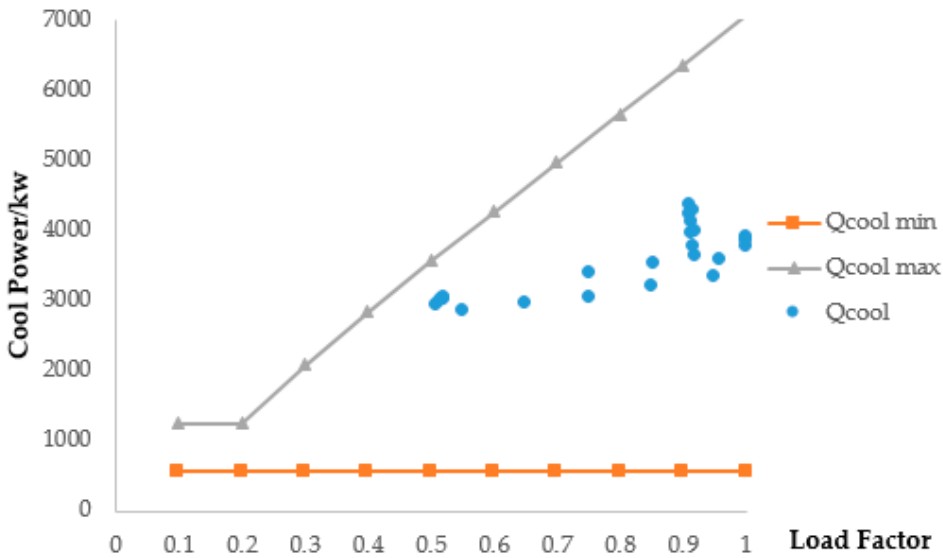

**Figure 13.** Multi-time scale cooling load-load rate operation.

For summer with large demand for cold energy, due to the strong compatibility and output characteristics of the cold energy system, there is still a wide range of load rate operation range selection

under the condition of large fluctuation of cold load throughout the day, so that the load rate operation point can be located inside the output range and the cold energy subsystem can operate stably.

To sum up, under the condition that the cold/heat/electric loads are independent of each other and do not interfere with each other before summer, according to the analysis of the load rate operation results of each decoupling subsystem, the "source-load" power of the system is balanced and stable in operation.

### 5.3. Comparison between Multi-Time Scale and Single-Time Scale Systems

In order to analyze the difference of time complementarity between systems, this paper compares multi-time scale systems with single time scale systems. In a single-time scale system, according to the principle of minimum adaptability of load scheduling time, i.e., taking the longest scheduling period as the scheduling time, the cold/heat/electric energy systems will be unified into the same time scale scheduling ($\Delta t_1 = 1$ h), the multi-time scale electric energy systems will be unified into one hour by 15 min scheduling, the heat energy systems will be unified into one hour by 30 min scheduling, and the original load parameters will be added into one hour to calculate. As the peak power load of a single-time scale system increases nearly 4 times as much as that of a multi-time scale system, and the gas internal combustion engines cannot match the applicable models due to capacity, power and other reasons, it is necessary to increase the number of gas internal combustion engines to 4 in a single-time scale system and keep the load rates of each gas internal combustion engine consistent so as to facilitate systematic analysis and comparison. Then the single time scale system operation results are shown in the following Figures 14–16:

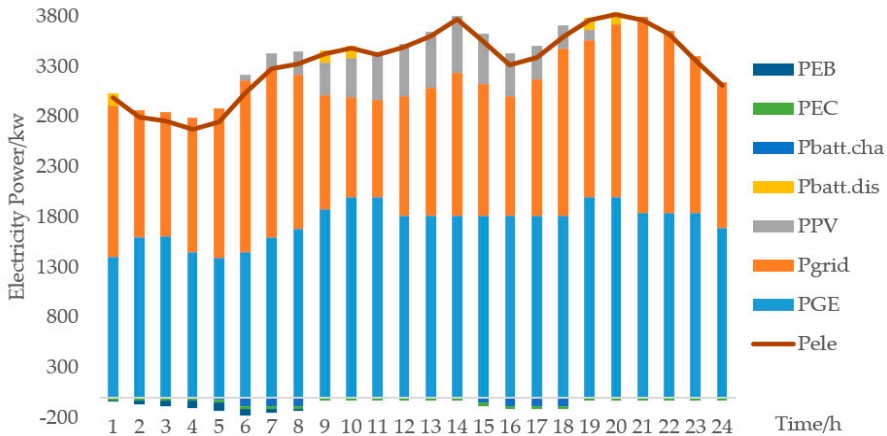

**Figure 14.** Power output diagram of various equipment before single time scale in summer.

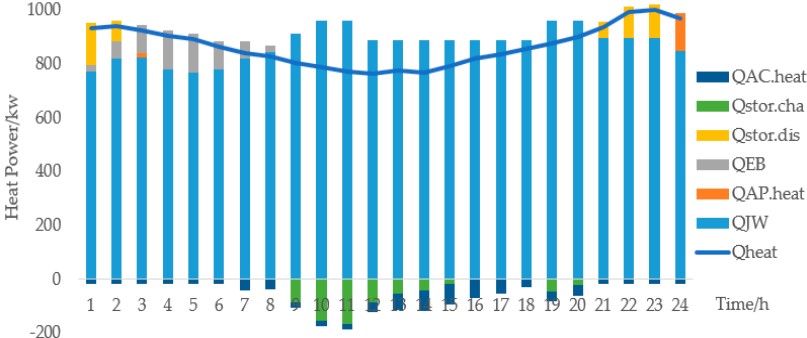

**Figure 15.** Heat output diagram of various equipment before single time scale in summer.

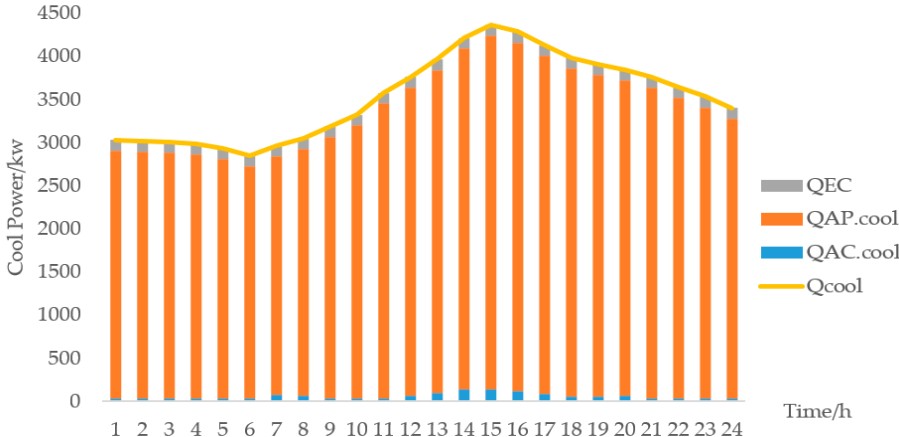

**Figure 16.** Cooling output diagram of various equipment before single time scale in summer.

The power output of each equipment before the single time scale in summer is shown in Figure 14. The gas internal combustion engine operates at a load rate of more than 80% throughout the day, taking up half of the power load throughout the day. The system has completely absorbed the photovoltaic power and greatly reduced the dependence on the grid power when the system is running at noon (the 10th to 15th hour). Electric boilers and electric refrigerators absorb only a small part of electric energy throughout the day. However, power storage equipment is put into operation at certain times to maintain stable operation of the power system.

The heat energy output of each equipment before the single time scale in summer is shown in Figure 15, and the cylinder liner water heat exchanger outputs sufficient thermal energy. Electric boiler and flue gas absorption heat pump output less; The absorption refrigerator only absorbs a small amount of heat energy and converts it into cold energy. During the first 2, 9, 15 and 19, 23 h of system operation, the heat storage equipment absorbs and releases power to maintain the stability of the thermal system.

The cooling energy output of each equipment before the single time scale in summer is shown in Figure 16. The flue gas absorption heat pump provides almost all-day cooling energy load, while the electric refrigerator and absorption refrigerator are only used as auxiliary equipment, thus the cooling energy power of the system is balanced.

In terms of single-time scale system load rate operation, the relationship between system load and operation load rate is shown in the following Figures 17–19:

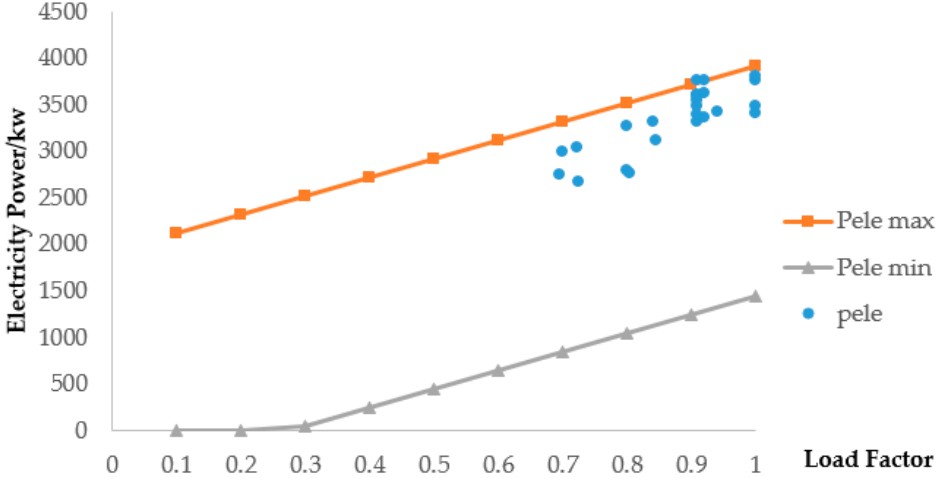

**Figure 17.** Single time scale power load-load rate operation.

Single-time scale power load-load ratio operation is shown in Figure 17. Although most power load ratio operation points are close to the upper limit of power output of the decoupled power system and the power system operates efficiently, some power load ratio operation points exceed the upper limit of power output boundary. At this time, the power storage equipment is put into operation to absorb some power, otherwise the system cannot operate stably, and at the same time, the energy storage burden of the system is increased.

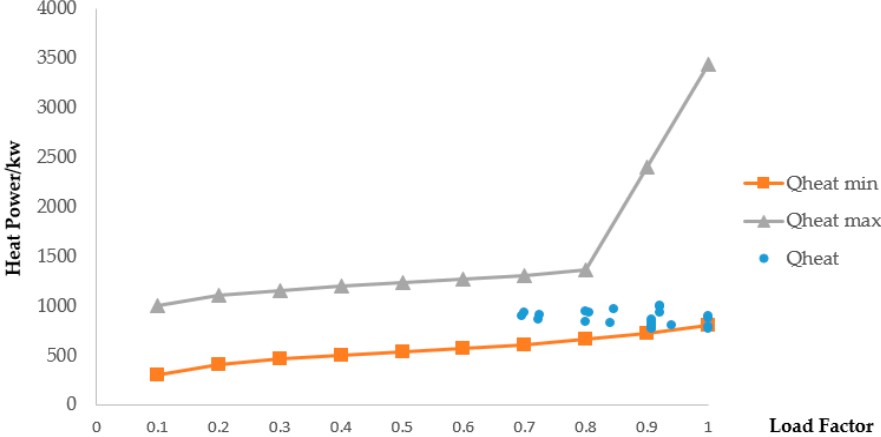

**Figure 18.** Single time scale heat load-load rate operation.

Single-time scale heat load-load ratio operation is shown in Figure 18. summer heat load is basically linearly distributed. Due to single-time scale adjustment, the heat load of the system is almost twice that of multiple time scales within one hour. At some operating points with high load rate, the lower limit of the power output boundary has been dropped. At this time, the heat storage equipment must be operated to ensure the stability of the system output heat energy.

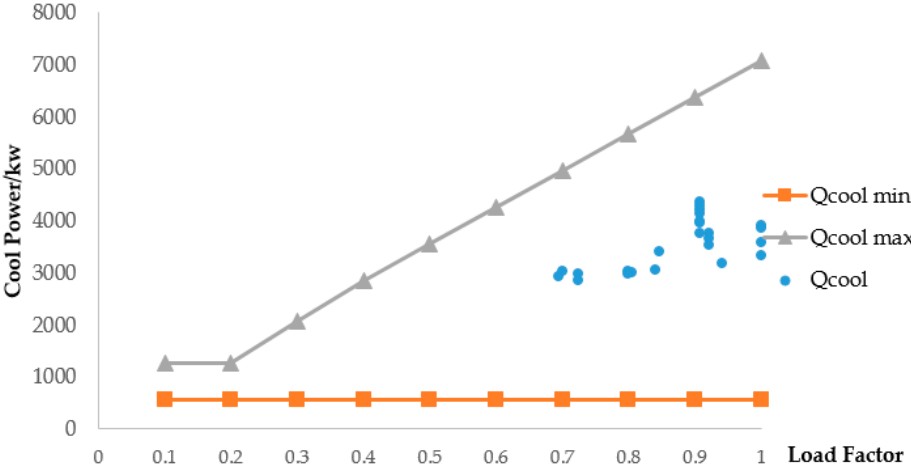

**Figure 19.** Single time scale cooling load-load rate operation.

The operation of single-time scale cooling load-load ratio is shown in Figure 19. Although the cooling energy scheduling period of single-time scale system is the same as that of multi-time scale system, the system load ratio is determined by the loads of cold, heat and electricity, which makes the overall load ratio of single-time scale system higher and the system is inferior to multi-time scale system in load ratio selectivity.

Comparing the load rates of single-time scale system and multi-time scale system, it is found that the load rates of single-time scale system are kept above 80% throughout the day, while the load rates of

multi-time scale system are scattered, ranging from 55% to 100%. On the surface, the single-time scale system load rate is more stable and efficient, but in fact, the single-time scale system maintains a high load rate and the system operates at high load, which will greatly increase the system equipment loss and maintenance cost, which is very unfavorable to the long-term stable operation of the system and also increases the risk of system failure. The load rate of multi-time scale system is stable at 55~70% in low load period, but it can reach full load operation in high load period. The feasible range of system load rate is wider and the system has wider operation selection space.

A comparison of system operation results on multiple time scales and on a single time scale is shown in Table 2 below.

**Table 2.** Comparison table of results for multi-time scale and single-time scale systems.

|  | $F_{\text{run}}$ (CNY) | $F_{\text{poll}}$ (m$^3$) |
| --- | --- | --- |
| multiple time scales | 46,771.92 | 121.84 |
| single time scale | 54,068.34 | 149.01 |

From Table 2, it can be seen that the results of the multi-time scale system are better than those of the single time scale system in terms of total operating costs and pollutant gas emissions, with a reduction of 15.6% in economic operating costs and 22.3% in pollutant gas emissions. Under the condition of multi-time scale, the system equipment has more flexible time collocation and energy complementary selection, highlighting the multi-energy complementary characteristics of the cold-heat-electricity integrated energy system. However, under the condition of a single time scale, the flexible multi-time scale adjustment mode of the power system and the thermal system is abandoned, and the load burden of the system is increased in the same time dimension, so that the system has to expand the upper limit of equipment capacity or increase the equipment investment when selecting equipment, which is extremely dependent on the energy storage equipment to maintain the "source-load" power balance of the system, thus greatly increasing the investment cost of the system, the difficulty in equipment selection and the maintenance cost of the equipment, and the multi-time scale system has a wider operating load rate range.

## 6. Summary

The cold-thermal-electricity integrated energy system takes coupling multiple energy production modes to realize multi-energy complementation, energy step utilization and overall energy utilization rate improvement as the core, thus achieving the purposes of high efficiency, energy conservation, environmental friendliness and compatibility. Because it is close to the energy consumption side, the energy loss in the energy transmission process is greatly reduced. The mutual supplement of various energy sources also reduces the losses between energy production at all levels. At the same time, there are still many difficulties in optimizing the integrated energy system. Based on the multi-objective and multi-time scale optimization problem of the cold-thermal-electricity integrated energy system under the condition of the feasible load rate interval, this paper studies the cold-thermal-electricity integrated energy system topology structure, takes the gas internal combustion engine and the flue gas absorption heat pump as the core, considers the feasible load rate interval of the system under the condition of strong coupling, and adopts the method of decoupling analysis of the system to analyze the cold, heat and electrolysis coupling subsystems. The analysis results show that the selection of the system load rate will directly determine the "source-load" power balance of the system. In order to ensure the stable operation of the system under the condition of independent and uncertain loads, the equipment parameters should be adjusted according to the fluctuation range of each load to ensure the operating conditions of the system, so that the system load rate can fall within the stable load rate range of the cold, heat and electrolytic coupling subsystems. On the issue of multi-objective and multi-time scale optimization, the multi-objective function takes into account the lowest system economic operation cost and the lowest pollutant gas emission. At the same time, considering the

differences of different energy attributes and energy scheduling characteristics, different time scales are selected to model the equipment model and the power model. The power control is sensitive and the scheduling is fast, with 15 min as an operation cycle. The flexibility of heat energy is obvious, taking 30 min as an operation cycle; Cold energy changes slowly, taking 1 hour as an operation cycle. In the aspect of multi-objective problem solving, the unitary linear weighting method is adopted to convert the multi-objective problem into a single-objective problem, and different weighting coefficients are set to optimize the solution, and the above mixed integer nonlinear problem is calculated by LINGO solver. This paper selects a hotel in Zhangjiakou, northern China, for simulation analysis of summer front cold, heat and electricity loads, and compares it with a single time scale system. The results show that the multi-time scale system reduces the economic operation cost by 15.6% and the pollution gas emission by 22.3% compared with the single time scale system. Under the multi-time scale condition, the system equipment has a wider load rate operation range, flexible time allocation and complementary energy selection, and greatly reduces the equipment capacity, investment cost, equipment selection difficulty and equipment maintenance cost of the single time scale system, which is more conducive to the stable operation of the system.

This study uses a static model. When considering multi-time scale optimization, the model parameters change dynamically. By the way, the optimization time of cold energy is long, so it can be considered to reduce the optimized time scale of cold energy appropriately. Although the optimal weight coefficient ($k_{run}$, $k_{poll}$) is selected after the system optimization results, the results show that when the weight coefficient ($k_{run}$, $k_{poll}$) is (0.9, 0.1), it performs best, and the target has little impact on the pollution emission control, which is one of the directions that this method can improve. In terms of future work, the system model can be optimized to improve the dynamic parameter changes of equipment, so as to build a real-time coordination and optimization system and improve the stability of comprehensive energy system.

**Author Contributions:** Methodology, B.O., Z.Y. and L.Q.; Writing—Original Draft Preparation B.O.; Guidance and Editing, Z.Y., C.L., L.Q. and D.L.

**Funding:** This work supported by "National Key Research and Development Program of China 2018YFB0905105".

**Acknowledgments:** The research gratefully acknowledge the LINGO software to solving optimization.

**Conflicts of Interest:** The authors declare no conflict of interest.

## Nomenclature

**Abbreviations**

| | |
|---|---|
| AC | absorption refrigerator |
| AP | smoke absorption heat pump |
| EB | electric boiler |
| EC | electric refrigerator |
| GE | gas engine |
| HS | heat storage equipment |
| IES | integrated energy system |
| JW | cylinder liner water heat exchanger |
| PV | photovoltaic generator set |
| ST | electricity storage equipment |

**Parameters and Variables**

| | |
|---|---|
| $a$ | the system load factor |
| $a_3$, $a_2$, $a_1$, $a_0$ | fitting constant of gas internal combustion engine |
| $b_5$, $b_4$, $b_3$, $b_2$, $b_1$, $b_0$ | fitting constant of flue gas absorption heat pump |
| $B_{stor}(t_1, t_{2,j})$ | real-time capacity of heat storage equipment |
| $B_{stor.max}$ | maximum capacity constraint of heat storage equipment, 720 KW |
| $B_{stor.min}$ | minimum capacity constraint for heat storage equipment, 160 KW |
| $COP_{AC}$ | energy efficiency coefficient of absorption refrigerator, 1.69 |

| | |
|---|---|
| $COP_{AP}(t_1,t_{2,i})$ | energy efficiency coefficient of flue gas absorption heat pump |
| $COP_{EB}$ | energy-producing coefficient of electric boiler, 1.8 |
| $COP_{EC}$ | energy efficiency coefficient of electric refrigerator, 4.1 |
| $C_W(t_1,t_{2,i})$ | specific heat capacity of hot water at different temperatures |
| $D_{batt.cha}(t_1,t_{2,i})$ | charging variables of power storage equipment |
| $D_{batt.dis}(t_1,t_{2,i})$ | discharge variables of electric storage equipment |
| $D_{stor.cha}(t_1,t_{2,j})$ | heat absorption variables of heat storage equipment |
| $D_{stor.dis}(t_1,t_{2,j})$ | heat release variables of heat storage equipment |
| $E_{batt}(t_1,t_{2,i})$ | real-time capacity of power storage equipment |
| $E_{batt.max}$ | maximum storage capacity of power storage equipment, 400 KW |
| $E_{batt.min}$ | minimum storage capacity of power storage equipment, 100 KW |
| $f_{gas}(t_1,t_{2,i})$ | natural gas price, 1.5 (CNY/m3) |
| $f_{grid}(t_1,t_{2,i})$ | real-time electricity price of power grid |
| $F$ | the mixed objective function value |
| $F_{gas}(t_1,t_{2,i})$ | cost of system purchase of natural gas |
| $F_{grid}(t_1,t_{2,i})$ | electricity Purchase Expenses for System and Power Grid |
| $F_{run}$ | the total operating cost of the system |
| $F_{run.min}$ | the minimum optimal values of economic operation |
| $F_{main}(t_1)$ | maintenance cost of system equipment |
| $F_{poll}$ | emissions of polluting gases |
| $F_{poll.min}$ | the minimum optimal values of economic operation pollutant gas emission |
| $G_{STC}$ | rated irradiation intensity of photovoltaic generator sets |
| $G_{ING}(t_1,t_{2,i})$ | real-time irradiation intensity |
| $k$ | generation coefficient of photovoltaic generator set, −0.0047% |
| $k_L$ | self-loss coefficient of power storage equipment, 0.04 |
| $k_s$ | self-loss coefficient of heat storage equipment, 0.02 |
| $k_{run}$ | the economic operation weight coefficient |
| $k_{poll}$ | the weight coefficient of pollutant gas emission |
| $k_{AC.heat}[Q_{AC.heat}(t_1,t_{2,j})]$ | maintenance coefficient of absorption refrigerator, 0.02 |
| $k_{AP.cool}[Q_{AP.cool}(t_1,t_{2,i})]$ | cold power maintenance coefficient of flue gas absorption heat pump, 0.01 |
| $k_{AP.heat}[Q_{AP.heat}(t_1,t_{2,i})]$ | thermal power maintenance coefficient of flue gas absorption heat pump, 0.01 |
| $k_{batt.dis/cha}[P_{batt.dis/cha}(t_1,t_{2,i})]$ | power maintenance coefficient of power storage equipment, 0.02 |
| $k_{GE}[P_{GE}(t_1,t_{2,i})]$ | maintenance coefficient of gas internal combustion engine under different output power |
| $k_{PV}[P_{PV}(t_1,t_{2,i})]$ | maintenance coefficient of photovoltaic generator set, 0.01 |
| $k_{stor.dis/cha}[Q_{stor.dis/cha}(t_1,t_{2,j})]$ | power maintenance coefficient of heat storage equipment, 0.015 |
| $L_{cool}(t_1,t_{2,i})$ | cold water flow rate of flue gas absorption heat pump |
| $L_{cool.max}$ | maximum cooling flow rate of flue gas absorption heat pump, 8 L/h |
| $L_{heat}(t_1,t_{2,i})$ | hot water flow rate of flue gas absorption heat pump |
| $L_{heat.max}$ | maximum heating flow of flue gas absorption heat pump, 10 L/h |
| $LHV$ | low calorific value of natural gas, 9.7 KW/m$^3$ |
| $P_{batt.cha}(t_1,t_{2,i})$ | charging power of power storage equipment |
| $P_{batt.cha.max}$ | maximum charging power of power storage equipment, 120 KW |
| $P_{batt.cha.min}$ | minimum charging power of power storage equipment, 0 KW |
| $P_{batt.dis}(t_1,t_{2,i})$ | discharge power of power storage equipment |
| $P_{batt.dis.max}$ | maximum discharge power of power storage equipment, 90 KW |
| $P_{batt.dis.min}$ | minimum discharge power of power storage equipment, 0 KW |
| $P_{batt.dis/cha}(t_1,t_{2,i})$ | interactive power of power storage equipment |
| $P_{ele}(t_1,t_{2,i})$ | electrical load |
| $P_{EB}(t_1,t_{2,i})$ | electric boiler input power |
| $P_{EB.max}$ | maximum electric power of electric boiler, 80 KW |
| $P_{EB.min}$ | minimum electric power of electric boiler, 0 KW |
| $P_{EC}(t_1,t_{2,i})$ | input electric power of electric refrigerator |

| | |
|---|---|
| $P_{EC.\max}$ | maximum electric power of electric refrigerator, 60 KW |
| $P_{EC.\min}$ | minimum electric power of electric refrigerator, 30 KW |
| $P_{grid}(t_1,t_{2,i})$ | system and Power Grid Power Purchase |
| $P_{GE}(t_1,t_{2,i})$ | output electric power of gas internal combustion engine |
| $P_{GE.\max}$ | the output gradient of gas internal combustion engine is limited to 50 KW |
| $P_{\max}$ | the rated power of gas internal combustion engine is 500 KW. |
| $P_{PV}(t_1,t_{2,i})$ | power generation of photovoltaic generator sets |
| $P_{STC}$ | rated output of photovoltaic generator Sets |
| $Q_{AC.cool}(t_1)$ | cold power output by absorption refrigerator |
| $Q_{AC.cool.\max}$ | output gradient constraint of absorption refrigerator, 40 KW |
| $Q_{AC.heat}(t_1,t_{2,j})$ | heat power absorbed by absorption refrigerator |
| $Q_{AC.heat.\max}$ | maximum thermal power absorbed by absorption refrigerator, 80 KW |
| $Q_{AC.heat.\min}$ | minimum thermal power absorbed by absorption refrigerator, 20 KW |
| $Q_{AP.cool}(t_1,t_{2,i})$ | smoke absorption heat pump outputs cold power |
| $Q_{AP.cool.\max}$ | cooling power output gradient constraint of flue gas absorption heat pump, 500 kw |
| $Q_{AP.heat}(t_1,t_{2,i})$ | smoke absorption heat pump outputs heat power |
| $Q_{AP.heat.\max}$ | heating power output gradient constraint of flue gas absorption heat pump, 400 kw |
| $Q_{cool}(t_1)$ | cold load |
| $Q_{EB}(t_1,t_{2,j})$ | electric boiler output thermal power |
| $Q_{EB.\max}$ | output slope constraints of electric boilers |
| $Q_{EC}(t_1)$ | output cooling power of electric refrigerator |
| $Q_{EC.\max}$ | output gradient constraint of electric refrigerator, 60 KW |
| $Q_{heat}(t_1,t_{2,j})$ | thermal load |
| $Q_{JW}(t_1,t_{2,i})$ | output thermal power of cylinder liner water heat exchanger |
| $Q_{stor.cha}(t_1,t_{2,j})$ | heat absorption power of heat storage equipment |
| $Q_{stor.cha.\max}$ | maximum heat absorption power of heat storage equipment, 200 KW |
| $Q_{stor.cha.\min}$ | minimum heat absorption power of heat storage equipment, 0 KW |
| $Q_{stor.dis}(t_1,t_{2,j})$ | heat release power of heat storage equipment |
| $Q_{stor.dis.\max}$ | maximum heat release power of heat storage equipment, 250 KW |
| $Q_{stor.dis.\min}$ | minimum heat release power of heat storage equipment, 0 KW |
| $Q_{stor.dis/cha}(t_1,t_{2,j})$ | interactive power of heat storage equipment |
| $t_1$ | hours of operation in a day |
| $t_2$ | minutes in an hour |
| $\Delta t_1$ | scheduling period of cold energy |
| $\Delta t_{2,i}$ | scheduling period of electric energy |
| $\Delta t_{2,j}$ | scheduling period of heat energy |
| $T(t_1,t_{2,i})$ | inlet temperature of flue gas absorption heat pump |
| $T_{cool}$ | cold water outlet temperature, 40 °C |
| $T_{heat}$ | hot water outlet temperature, 100 °C |
| $T_{out}(t_1,t_{2,i})$ | ambient temperature |
| $T_s$ | reference temperature of generator set, 25 °C |
| $V_{gas}(t_1,t_{2,i})$ | the system consumes natural gas volume |
| $\alpha_{sour}$ | pollution gas emission coefficient at power supply side of power grid, 0.0009 |
| $\alpha_{tran}$ | emission coefficient of polluted gas transported by power grid lines, 0.000825 |
| $\alpha_{PGE}$ | pollution gas emission coefficient of gas internal combustion engine, 0.0015 |
| $\lambda_{heat}(t_1,t_{2,i})$ | smoke heating ratio of smoke absorption heat pump |
| $\lambda_{cool}(t_1,t_{2,i})$ | smoke refrigeration ratio of smoke absorption heat pump |
| $\eta_{GE.elec}(t_1,t_{2,i})$ | power generation efficiency of gas internal combustion engine |
| $\eta_L$ | natural loss rate of gas internal combustion engine, 0.08 |
| $\eta_{gas}$ | natural gas utilization rate of gas internal combustion engine, 0.98 |
| $\eta_{AP.heat}$ | thermal efficiency of flue gas absorption heat pump, 0.62 |

| | |
|---|---|
| $\eta_{AP.cool}$ | refrigeration efficiency of flue gas absorption heat pump, 0.58 |
| $\eta_{JW}$ | heat transfer efficiency of cylinder liner water heat exchanger, 0.2 |
| $\eta_{batt.cha}$ | charging efficiency of power storage equipment, 0.95 |
| $\eta_{batt.dis}$ | discharge efficiency of power storage equipment, 0.95 |
| $\eta_{stor.cha}$ | heat absorption efficiency of heat storage equipment, 0.98 |
| $\eta_{stor.dis}$ | heat release efficiency of heat storage equipment, 0.98 |

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
