# Peer review of "Research on Multi-Time Scale Optimization Strategy of Cold-Thermal-Electric Integrated Energy System Considering Feasible Interval of System Load Rate"

_energies, doi:10.3390/en12173233_

Round 1

Reviewer 1 Report

I have three major concerns about the optimizing process (it is called  "model solving process" in the manuscript, which I don't think is correct as it should be a optimization problem):

In Figure 3, when the constraints are not satisfied, why the arrow goes to "calculate the output of each equipment"? shouldn't be re-selecting the weights or the input parameters? How do you make sure the objective function is the "global optimal solution"? Also, is it optimal respect to the weights or the input parameters or both? How do you choose the weights, how many pairs do you need to choose and why?

Author Response

Dear reviewer

      This is my revised manuscript, please check it!

It's my pleasure

                                                                     Bin . Ouyang .

Reviewer 2 Report

The comments that need to be addressed are summarized in the pdf attached. The comments are referenced as much as possible using line numbers and sections. 

Author Response

(The authors gave the same response as above.)

Round 2

Reviewer 2 Report

Most of the comments have been addressed. Following issues still need some review.

1. Careful proofread will be required for the edited sections to improve the language. Several typos and grammatical errors are in the edited parts.

2. The explanation in regards to the comments for weight coefficients are still not satisfactory. The comment was "The crux of multi-objective optimization is the weights given to its different competing objectives. This makes the problem interesting, since many times you cannot compare the two entities as is the case here. The values of pollution and costs is hard to compare. This can be circumvented by giving a cost factor to pollution. I would suggest the author to do that and compare results if possible. The other way is to give solution as a Pareto frontier which gives the power to the decision maker to choose the right weights. Otherwise, please put it in the future work and shortcoming of this model in the conclusion section. I can see that authors have tried to normalize the objectives. However, we are still comparing costs and pollution. The main problem I see is that the weights taken here put very less emphasis on minimizing the pollution, for which the question arises, the purpose of the model was supposed to look pollution as well. However, this can be mentioned in the conclusions sections as the future work or improvement in the model to consider the pollution index." The comment is the values chosen is very biased towards economic operation. I would suggest the authors to explain this as a note in conclusions and future work or explain is it appropriate to take the coefficient so small for pollution objective.

3. Secondly, for finding the weight coefficient, authors have varied the coefficient to get the coefficients at which the combined objective has maximum value. Author should explain at what conditions these coefficients were found. Will changing other parameters of the model change the coefficients? This needs to be explained as mentioned in the last review. 

4. Please delete the line "The following problems are found in this study: on the one hand, the model is defective." in the conclusions. It seems inappropriate. 

Author Response

Dear Review:

      Thank you very much for receiving your opinion. I have revised it based on your opinion. Please see the attachment for the detailed content: Response to Reviewer 2 Comments (Round2).

Best wishes

                                                                                           Bin Ouyang.     
